# Smartphone Sensors for Health Monitoring and Diagnosis

**DOI:** 10.3390/s19092164

**Published:** 2019-05-09

**Authors:** Sumit Majumder, M. Jamal Deen

**Affiliations:** 1Department of Electrical and Computer Engineering, McMaster University, Hamilton, ON L8S 4L8, Canada; majums3@mcmaster.ca; 2School of Biomedical Engineering, McMaster University, Hamilton, ON L8S 4L8, Canada

**Keywords:** smartphone, remote healthcare, mHealth, telehealth, medical device, regulation, smartphone sensor

## Abstract

Over the past few decades, we have witnessed a dramatic rise in life expectancy owing to significant advances in medical science and technology, medicine as well as increased awareness about nutrition, education, and environmental and personal hygiene. Consequently, the elderly population in many countries are expected to rise rapidly in the coming years. A rapidly rising elderly demographics is expected to adversely affect the socioeconomic systems of many nations in terms of costs associated with their healthcare and wellbeing. In addition, diseases related to the cardiovascular system, eye, respiratory system, skin and mental health are widespread globally. However, most of these diseases can be avoided and/or properly managed through continuous monitoring. In order to enable continuous health monitoring as well as to serve growing healthcare needs; affordable, non-invasive and easy-to-use healthcare solutions are critical. The ever-increasing penetration of smartphones, coupled with embedded sensors and modern communication technologies, make it an attractive technology for enabling continuous and remote monitoring of an individual’s health and wellbeing with negligible additional costs. In this paper, we present a comprehensive review of the state-of-the-art research and developments in smartphone-sensor based healthcare technologies. A discussion on regulatory policies for medical devices and their implications in smartphone-based healthcare systems is presented. Finally, some future research perspectives and concerns regarding smartphone-based healthcare systems are described.

## 1. Introduction

Life expectancy in many countries has increased drastically over the last several decades. This large increase can be attributed primarily to the remarkable advances in healthcare and medical technologies, and the growing consciousness about health, nutrition, sanitation, and education [1,2,3,4]. However, this increased life expectancy, combined with the globally decreasing birthrate is expected to result in a large aging population in the near future. In fact, the elderly population over the age of 65 years is expected to outnumber the children under the age of 14 years by 2050 [3]. Furthermore, approximately 15% of the world’s population have some forms of disability, and 110–190 million adults suffer from major functional difficulties [5]. Disability of any form in a person limits mobility and independence, thus preventing or delaying them from receiving necessary healthcare support on time. In addition, a significant number of people around the globe suffer from chronic diseases and medical conditions such as cardiovascular diseases, lung diseases, different forms of cancer, diabetes and diabetes-related complications. It is reported that six of ten American adults (>18 years) suffer from at least one chronic disease, with four of ten having multiple chronic conditions [6]. Further, chronic diseases account for ~65–70% of total mortality among the ten leading causes of death [7]. In fact, heart disease and cancer together account for 48% of all deaths, thus becoming the leading cause of mortality [7,8]. 

In addition, the unregulated blood sugar i.e. diabetes is likely to be the seventh leading cause of death by 2030 [9]. Diabetes increases the risk of long-term complications such as kidney failure, limb amputations, and diabetic retinopathy (DR). Diabetic retinopathy is an eye disease that results from the damage of retinal blood vessels due to the prolonged presence of excessive glucose in the blood. It may lead to blindness if not treated in time. In fact, DR was estimated to account for 5 million blindness globally in 2002 [10]. Other prevalent eye-related diseases include cataract, glaucoma, and age-related macular degeneration (AMD) that together with DR caused 65% of all blindness globally in 2010, with cataract alone accounting for 51% [11]. Furthermore, poor air quality in many large cities threatens city-dwellers with diseases like asthma and lung diseases. Globally, an estimated 235 million people are currently suffering from asthma, which caused 383,000 deaths in 2015 [12]. Therefore, the demand for healthcare services is understandably rising more rapidly than ever before.

An important issue related to providing adequate healthcare services is the continuously increasing cost of pharmaceuticals, modern medical diagnostic procedures and in-facility care services, which together renders the existing healthcare services unaffordable. To give one example, in the 2017 budget of the Province of Ontario in Canada, an additional $11.5 billion was allocated for the next three years in healthcare sectors [13]. Further, the total health spending per Canadian was expected to be $6839 in 2018, representing more than 11% of Canada’s GDP and these numbers are similar to most other OECD (Organization for Economic Co-operation and Development) countries [14]. Therefore, present-day healthcare services are likely to cause a substantial socioeconomic burden on many nations, particularly the developing and least developed ones [15,16,17,18,19]. Furthermore, a large fraction of the elderly relies on other persons such as family members, friends and volunteers, or an expensive formal care services such as caregivers and elderly care centers for their daily living and healthcare needs [20,21,22]. Therefore, enabling superior healthcare and monitoring services at an affordable price is urgently needed, particularly for persons having limited access to healthcare facilities or to those living under constrained or fixed budget conditions. However, through long-term monitoring of key physiological parameters and activities of the elderly in a continuous fashion, many of the medical complications can be avoided or managed properly [22,23,24,25]. Long term monitoring of health enables early diagnoses of developing diseases. However, current practice requires frequent visits to or long term stays at expensive healthcare facilities. In addition, a shortage of skilled healthcare personnel, and limited financial capability, coupled with increasing healthcare costs [26] contribute to the bottleneck in realizing long-term health monitoring. Smartphone-based healthcare systems, on the other hand, can potentially enable a cost-effective alternative for long-term health monitoring and may allow the healthcare personnel to monitor and assess their patients remotely without interfering with their daily activities [22,27].

The enormous advances in energy efficient and high-speed computing and communication technologies have revolutionized the global telecom industry. Furthermore, the significant progress in display, sensor and battery technologies together have paved the way for modern mobile devices such as smartphones and tablets, enabling seamless internet connectivity, entertainment, and health and fitness monitoring on the go along with conventional voice and text communication. Smartphones have grown in popularity over the past decade and by 2021, the global penetration of smartphones is expected to exceed 3.8 billion [28]. Modern day smartphones come with a number of embedded sensors such as a high-resolution complementary metal-oxide semiconductor (CMOS) image sensor, global positioning system (GPS) sensor, accelerometer, gyroscope, magnetometer, ambient light sensor and microphone. These sensors can be used to measure several health parameters such as heart rate (HR), HR variability (HRV), respiratory rate (RR), and health conditions such as skin diseases and eye diseases, thus turning the communication device into a continuous and long-term health monitoring system. Table 1 presents the health parameters and conditions that can be monitored using current embedded sensors of the smartphone. The data that are measured by the sensors can be analyzed and displayed on the phone and/or transmitted to a distant healthcare facility or healthcare personnel for further investigation over the wireless mobile communication platform such as 3G, high speed packet access (HSPA), and long-term evolution (LTE). These existing platforms offer high-speed and seamless internet connectivity even on the go, thereby allowing people to remain connected with their healthcare providers [7,22].

In this article, we present a detailed review of the current state of research and development in the health monitoring systems based-on embedded sensors in smartphones. In Section 2, we discuss the evolution of smartphones. Some important recent works on different health monitoring systems are presented in Section 3, which is followed by a discussion (Section 4) on the regulatory policies associated with smartphone-based medical devices. Finally, the paper is concluded in Section 5 with a discussion on future research perspectives and some key challenges in realizing smartphone-based medical devices.

## 2. Evolution of Smartphone

In 1992, IBM announced a ground-breaking device named Simon Personal Communicator that brought together the functionalities of a cellular phone and a Personal Digital Assistant (PDA) [29,30,31], thus unveiling a whole new concept of so-called ‘smartphone’ in the cellular phone industry. Simon featured a 4.5” × 1.4” monochrome LCD touchscreen and came with a stylus and a charging base station. Along with conventional voice communication, the device was also capable of communicating emails, faxes, and pages—some features that were later attributed to smartphones. Simon featured a notes collection to write in, an address book, calendar, world clock and an appointment scheduler, and was also flexible to third-party applications [29,30,31]. While it was a giant leap into the market by IBM, it, however, was expensive, costing the customer $899 with a service contract. Being much ahead of its time and with such a high price tag, Simon failed to attract customers. Even though the tech giant sold approximately 50,000 units in 6 months [32], it opted out of making a second-generation Simon.

In 1996, Nokia revealed a clamshell phone, Nokia 9000 Communicator, which opened to a full QWERTY keyboard and physical navigation buttons flanking a monochrome LCD screen nearly as big as the device itself. It featured Web browsing capability on top of most of the features that IBM’s Simon offered. However, Ericsson first coined the term ‘Smart-phone’ for its Ericsson GS 88. The GS 88, also known as ‘Penelope’, was strikingly similar to the Nokia 9000. However, it was never released to the public, arguably because of its weight and poor battery quality [31]. Later in 2000, Ericsson first officially used the term ‘Smartphone’ for the Ericsson R380, which was much cheaper, smaller and lighter than the Nokia 9000. Ericsson R380 was the first device to use the mobile-specific Symbian operating system (OS) and only second to IBM’s Simon to have a touchscreen in a phone. R380 was one of the first few smartphones that used the wireless application protocol (WAP) for faster and smoother web browsing.

In 2002, Handspring and RIM released their first smartphones Treo-180 and Blackberry 5810 (5820 for Europe) in the market, respectively [29,30,31]. The Treo-180 brought the functionality of a phone, an email messaging device and a PDA together, and enabled the users to check the calendar while talking on the phone and to dial directly from the contact list. The Blackberry 5810 featured enterprise e-mail and instant messaging services, text messaging, and a WAP browser. However, this device lacked an integrated microphone, for which the user had to attach a headset to make or receive calls. Both the BlackBerry 5810 and the Treo-180 featured a large monochrome screen and a QWERTY keyboard like a PDA. However, unlike the BlackBerry 5810, the Treo-180 came in a clamshell format with a visible antenna and a hinged lid over the phone that flipped up to serve as the earpiece for phone conversations. Then RIM released Blackberry 6210, otherwise known as the ‘BlackBerry Quark’ in the following year, featuring a built-in microphone and speaker [30,31].

In 2000, both Samsung and Sharp introduced a camera phone in their respective local markets. In South Korea, Samsung released SCH-V200 that featured a 1.5″ TFT-LCD display and a 0.35-megapixel video graphics array (VGA) camera that could capture up to 20 images. A few months later, in Japan, Sharp released the J-SH04 in Japan with a 256-color display and a built-in 0.11-megapixel CMOS camera. Although the camera resolution of J-SH04 was much less than that of the SCH-V200, it featured a phone-integrated camera in the true sense for the first time and allowed for transferring of images directly from it, whereas the SCH-V200 brought two separate devices in one enclosure and therefore needed to transfer the pictures to a computer for sharing. However, none of the camera phones supported web browsing and email communication until Sanyo launched the first smartphone with a built-in camera in 2002. The Sanyo SCP-5300 came in a clamshell format and featured dual color displays, WAP browser, and an integrated 0.3-megapixel camera with short-range LED light sensor pro flash. It also had brightness and white balance control, self-timer, digital zoom, and several filter effects such as sepia, black-and-white, and negative colors. 

Fast forward to a decade later, modern smartphones featured a number of sensors such as high-resolution and high-speed CMOS image sensor, GPS sensor, accelerometer, gyroscope, magnetometer, ambient light sensor, microphone, and fingerprint sensor (Figure 1). In addition, the processing and data storage capability of current smartphones has improved significantly. Figure 2 shows the built-in sensors that most present-day smartphones possess.

## 3. Smartphone Sensors for Health Monitoring

As discussed earlier, the modern-day smartphones are fitted with a number of sensors. These sensors allow for active and/or passive sensing of several health parameters and health conditions. The data thus measured by the smartphone-sensors, sometimes coupled with information related to device usage such as call logs, app usage and short message service (SMS) patterns can provide valuable information of an individual’s physical and mental health over a long period of time. In this way, the smartphone can potentially be turned into a viable and cost-effective device for continuous health monitoring. In the following sections (Section 3.1, Section 3.2, Section 3.3, Section 3.4, Section 3.5, Section 3.6 and Section 3.7), we discuss how the smartphone can be used for heart, eye, skin, mental health and activity monitoring, respectively.

### 3.1. Cardiovascular Health Monitoring

Heart rate (HR) or pulse rate is one of the four ‘vital signs’ that is routinely monitored by physicians to diagnose heart-related diseases such as different types of arrhythmias [22,33]. HR and HR variability (HRV) are typically extracted from the Electrocardiogram (ECG) (Figure 3a). However, these systems, particularly the conventional 12-lead ECG systems are expensive, restrict user’s movement and require trained medical professionals to operate in clinical settings. HR and HRV can also be measured using portable and hand-held single-lead ECG devices [22,34]. Furthermore, with the advancement of wearable sensor technologies, HR and HRV can now be obtained using commercial fitness trackers such as Fitbit^®^ (San Francisco, CA, USA), Jawbone^®^ (San Francisco, CA, USA), Striiv^®^ (Redwood City, CA, USA), and Garmin^®^, (Olathe, KS, USA) [22]. 

However, these portable and wearable systems require additional accessories, which can be avoided by exploiting the embedded sensors such as a camera and microphone in the smartphone for monitoring HR and HRV. Using smartphone camera sensors, it is possible to estimate HR and HRV from the photoplethysmogram (PPG) signal derived from the video of the bare skin such as of the fingertip (Figure 3b) or the face. The light absorption characteristics of hemoglobin in blood differ from the surrounding body tissues such as flesh and bone. PPG estimates the volumetric changes in blood by detecting the fluctuation of transmissivity and/or reflectivity of light with arterial pulsation through the tissue (Figure 4) [22,35]. Although near-infrared (NIR), red light sources are used in most commercial systems [22,35], some researchers [36,37,38,39,40] exploited the smartphone embedded white flashlight to illuminate the tissue to measure the PPG.

Most published smartphone-based HR and HRV monitoring applications [36,37,38,39,40] follow a similar approach where these parameters are extracted from the PPG signal either by measuring pulse-to-pulse time difference in time-domain [39] or by finding the dominant frequency in the frequency domain [36,38]. In Reference [39], the HR and HRV were estimated from the PPG signal obtained from the fingertip using the flashlight and the camera of a smartphone. The green channel of the video data was used to derive the pulse signal after a low-frequency band-pass filtering. Instead of using a conventional peak/valley detection method, the authors detected the steepest slope of each pulse wave and evaluated the correlation of the PPG signal with a pulse wave pattern to determine the cardiac cycles. The calculated HR and HRV were highly correlated to that measured from a commercial ECG monitor, although the degree of improvement in the measurement accuracy with the proposed algorithm over the conventional method was not reported. In addition, the authors did not evaluate the performance of the proposed algorithm when there were any bodily movements. A previously reported [41,42] motion detection technique was employed in Reference [36] to identify and discard the corrupt video data that is highly affected by motion artifacts. The PPG signal was derived from the video of the index fingertip recorded using all three channels (red, blue and green) of a smartphone camera. They calculated a threshold based on the difference of the maximum and the minimum intensity and summed up the pixel values with intensity greater than the threshold for each frame, from which the PPG signal was obtained. The periodic change in the blood volume flow during the cardiac cycles was reflected in the PPG signal acquired through the red channel in comparison to the other two channels. The pulse rate was finally extracted by performing simple Fast Fourier Transform (FFT) analysis on the PPG obtained through the red channel, achieving an average accuracy of as high as 98% with a maximum error of three beats per minute (bpm) with respect to the actual pulse. However, the algorithm cannot correct motion artifact in the video. Rather, it relies on restarting the video recording when the movement exceeds a predefined level. 

In Reference [38], both the front- and rear-cameras of a smartphone were used to simultaneously monitor the heart rate (HR) and respiration rate (RR). The HR was obtained as usual from the PPG signal of the fingertip placed on top of the rear-facing camera. The front camera was used to estimate the RR by detecting the movement of the chest and the abdomen. The HR and RR were obtained by identifying the dominant frequency of the images in frequency-domain (Welch’s power spectral density). The authors reported achieving a high (95%) agreement in the measured HR compared to that obtained using the standard ECG from 11 healthy subjects with a variation ranging from −5.6 to 5.5 beats-per-minute. The RR estimated from the chest and the abdominal walls resulted in average median errors of 1.4% and 1.6%, respectively. In order to achieve a wide dynamic range of 6–60 breaths per minute, an automatic region-of-interest (ROI) selection protocol was used. With this protocol, a signal either from the abdomen or the chest based on the absolute value of mean autocorrelation was selected. However, this approach of monitoring RR restricts the movement of the body during video recording to avoid motion artifacts (MA). MA can be eliminated or reduced by incorporating MA estimation algorithms [41,42,43] in the system. In addition, extra caution is necessary while estimating RR in presence of colorful and patterned cloths, which can affect the fidelity of video recordings, thereby affecting the estimation accuracy of the RR. 

All HR monitoring systems discussed above are contract-based, which require the user to keep the fingertip in close contact with the smartphone camera lens using sufficient strength. Any alteration of the finger position and illumination condition may result in an erroneous estimate of HR [44]. Contactless monitoring systems, on the other hand, estimate HR from the PPG signal derived from the video of the face. Such a non-contact cardiac pulse monitoring application named FaceBeat was presented in Reference [45]. The application was based on an algorithm similar to the first-of-its-kind video-based HR monitoring system proposed in Reference [46], which had exploited the webcam of a laptop for video recording. FaceBeat extracts cardiac pulse and measures HR from the video of a user’s face recorded using the front camera of the smartphone. The photodetector array of the camera sensor detects the variation in the reflected light from a specific region of interest (ROI) in the face with the change of blood volume in the facial blood vessels. The authors exploited the green channel data of the recorded video, which as reported in References [40,44,47,48], is most suitable for evaluating HR, particularly in the presence of motion artifacts. The authors used independent component analysis (ICA) to remove noise and motion artifacts from the video data of the ROI and performed the frequency domain analysis on the both the raw and decomposed signals to extract HR and HRV. The HR thus measured showed a maximum average error of 1.5% with respect to the reference ECG signal obtained from a commercial ECG monitor. However, the complex computation process required for the application increase the processor load and computation time, thereby increasing the power consumption and reducing the battery life.

Unlike the conventional approach, which estimates HR from the fluctuation of reflected light through specific color channels (R, G, B), researchers in Reference [49] proposed a method that estimates both HR and RR by detecting the variation in the hue of the reflected light from the face. A 20 s video of the subject’s face was recorded. However, only the forehead region of the face was analyzed in the frequency domain to determine the dominant frequencies in the time-varying changes of the average Hue. The authors reported achieving highly accurate HR and RR measurements showing a higher correlation to the measurements with standard instruments than the green channel PPG. Nevertheless, this approach may not work if the forehead skin is covered with any object such as hair, headband or cap, or has scar tissue on it. Although face-based contactless monitoring of HR offers a more convenient alternative to the contact-based systems, its performance can vary with the variation in illumination, skin color, facial hair, scar and movement of the face. Table 2 presents some smartphone-sensor based cardiovascular monitoring systems presented in the literature.

### 3.2. Pulmonary Health Monitoring

Air pollution across the globe has increased significantly in the last decade [54] resulting in billions of people being at increased risk for chronic pulmonary diseases such as cough, asthma, chronic obstructive pulmonary disease (COPD) and lung cancer. In addition, smoking tobacco is one of the key risk factors for lung cancer and other pulmonary diseases [55]. In fact, lung cancer is the most common form of cancers and caused ~19% of all cancer-related deaths in 2018 [55]. Therefore, early detection of these lung diseases and continuous monitoring of pulmonary health are paramount for timely and effective medical intervention. Many researchers [56,57,58,59,60,61,62] used the microphone of a smartphone to detect the sound of a cough and breathing and analyzed the recorded audio signals in efforts to develop a cost-effective and portable tool for faster assessment of pulmonary health.

The smartphone can be used for lung rehabilitation exercise. In Reference [56], an interactive game was developed where the users play to dodge obstacles through inhalation and exhalation. The game, ‘Flappy Breath’ was designed for smartphones and either used its built-in microphone or a Bluetooth-enabled stretchable chest belt to detect breathing. When played using the microphone, the game detects the frequency and strength of the input sound, calculates the average volume of airflow, and also calibrates itself to identify the frequencies corresponding to silence, inhalation and exhalation. However, interferences from other nearby sources can corrupt the sound of breathing and may even make the game unresponsive to breathing. In contrast, the game does not require any initial calibration when played using the chest belt. It also allows the users to keep their hands free, but it incurs an additional cost to the user. The app can also store a person’s breathing patterns and information related to their airflow over a long period of time that can be useful in the long-term monitoring of lung health.

A cough detection algorithm, proposed in Reference [57], was used to analyze the audio signal recorded by the smartphone’s microphone to detect the cough event. There, the authors first obtained the spectrogram from the time-varying audio signal, and the cough sound was found to generate higher energy over a wide frequency range compared to other sounds such as throat clearing, speech, and noise. Based on the distinct pattern, the spectrogram of the cough sound from several training sequences was isolated, normalized and analyzed using the principal component analysis (PCA). A subset of the key principal components was then used to reconstruct the cough signal with high fidelity and finally, to classify the cough events. The authors reported achieving a sensitivity of 92% and a false positive rate of 0.5% in recognizing cough events using a random forest classifier. However, the phone was placed around the user’s neck or in the shirt pocket to improve the audio quality of the recorded sound, which may not be comfortable or always feasible for regular use. Therefore, further research is needed to implement the algorithm in a mobile platform for a complete phone-based cough detection system. A similar phone-based system for identifying and monitoring nasal symptoms such as blowing the nose, sneezing and runny nose was proposed in Reference [58]. This system also tracks the location information using the GPS data in the case of a nose related event, thereby allowing it to keep a record of contextual information related to the event for future reference. 

Some researchers [58,59,60,61,62] used the microphone of the smartphones to realize a low-cost spirometer in a mobile platform. Spirometers are widely used in clinical settings to quantify the flow and volume of air inspired and expired by the lungs during breathing. In standard spirometry, a volume-time (VT) curve (Figure 5a) and a flow-volume (FV) curve (Figure 5b) are obtained from forced expiratory flow. These curves are used to extract some clinically important parameters such as forced vital capacity (FVC), forced expiratory volume in 1 s (FEV1), peak expiratory flow (PEF), and the ratio FEV1 to FVC (FEV1/FVC), which are routinely used clinically to determine the degree of airflow obstruction in patients with pulmonary disorders such as asthma, COPD and cystic fibrosis. However, clinical spirometers are expensive and cost several thousands of dollars. Therefore, a smartphone-based spirometer can be a viable low-cost alternative to these expensive spirometers, particularly for the people in remote areas and in underdeveloped countries.

A smartphone-based spirometer, ‘SpiroSmart’, was proposed in Reference [58] using the built-in microphone to record the sound of forced exhalation and send the audio data to a remote server for analysis and parameter extraction. In the server, an algorithm to estimate the flow rate from the audio signal of forced exhalation was implemented. There, the authors first compensated for the loss of air pressure as the sound travels a distance from the mouthpiece to the microphone. Then, they estimated the air pressure at the opening of the mouth, which was converted to flow rate. This signal was further processed to extract a set of features from a window of 15 ms to approximate the flow rate over time. Finally, the spirometric parameters were estimated from the flow rate features following a bagged decision tree and mean square error-based regression techniques, and k-means clustering. In addition, the FV curve was estimated by regression using a combination of conditional random field (CRF) and a bagged decision tree. The authors reported achieving a median error of ~8% for the four spirometric parameters in reference to the measurements from a clinical spirometer. In a later work, the same research group proposed a call-in based system in References [60,61], where instead of recording the sound on the phone, users can directly call-in to record the audio of forced exhalation in a remote server. The system, named ‘SpiroCall’, runs a similar algorithm in the server as ‘SpiroSmart’ and estimated the spirometric parameters from the audio data. The transmission of exhalation sound over voice channels was found to have a negligible effect on the bandwidth and resolution of the signal as well as on the accuracy of the system, thus making the system suitable for remote monitoring of lung health. 

In the smartphone-based spirometer proposed in Reference [62], the authors attached a commercial mouthpiece with the smartphone using a custom-made 3D-printed adapter fabricated with polylactic acid (PLA) material. Following a down-sampling and subsequent filtering of the original audio signal, the authors performed a time-frequency analysis on the recorded signal using variable frequency complex demodulation (VFCDM). From VFCDM, they derived two curves—maximum power of each sample against accumulated maximum power and accumulated maximum power against time—which showed a similar pattern to the typical FV curve and VT curve, respectively. Finally, FVC, FEV1, and PEF were extracted from these curves. Although the individual parameters showed poor correlation with the reference, the FEV1/FVC or the Tiffeneau-Pinelli index, which is a key metric in diagnosing chronic obstructive pulmonary disease (COPD), was found to be highly correlated (r=0.8) to the reference measurements with a root mean squared error (RMSE) of ~5.5% and ~14.5% in healthy persons and COPD patients, respectively.

In References [63,64], a smartphone application, ‘LungScreen’ was developed to realize a personalized tool for lung cancer risk assessment. The application assesses the risk of lung cancer based on the user’s response to an interactive questionnaire that includes duration of tobacco use, occupational environment and family history of lung cancer. Based on this information, it instantly classifies the users into three risk groups—low, moderate and high. Using the location information from the GPS sensor of the smartphone, the users at high risk of lung cancer are referred and navigated to their nearest screening centers for further investigation. Although the app can be useful for initial and fast screening of lung cancer, the sensitivity and specificity of the app were not reported [63]. In addition, in Reference [64], 32 participants were found positive in a low-dose computed tomography (LDCT) screening among the 158 participants (Baranya County, Hungary) who were identified by the app being at high risk of non-small-cell lung carcinoma (NSCLC), the false negative rate (FNR) of the app was not reported. 

### 3.3. Ophthalmic Health Monitoring

Diabetic Retinopathy (DR) is one of the common complications of diabetes, which if diagnosed late and left untreated, can lead to blindness. Currently, the seven-field stereoscopic-dilated fundus photographs are considered as the ‘gold standard’ for diagnosing DR by the Early Treatment of Diabetic Retinopathy Study (ETDRS) group [65,66]. However, this protocol requires an expensive imaging system, specially-skilled photography personnel, and specialized processing and storage of films. Single-field digital fundus photography (Figure 6), although not as comprehensive as the seven-field stereoscopic-dilated fundus photography, can still serve as a screening tool for DR before a detailed ophthalmic evaluation and management [67]. Single-field digital fundus imaging is less expensive and more convenient in comparison to the standard seven-field stereoscopic ETDRS photography. However, this system is still costly with a price of the complete imaging system ranging from several thousand to ten thousand dollars [68], thus limiting its large-scale adoption for diagnosing DR and other eyes-related diseases, especially in developing and under-developed countries. However, the ever-increasing popularity and rapid technological advances of smartphone cameras, coupled with modern-day cloud-based image processing, storage and management services have paved the way for low-cost and efficient remote screening and diagnosis of ophthalmic diseases. This smartphone-based imaging technology can potentially be useful in inpatient consultations and emergency room visits [69].

At present, there are some smartphone applications available to perform simple ophthalmic tests, although their reliability and performance are often not guaranteed. In order to diagnose Diabetic Macular Edema (DME), Diabetic Retinopathy (DR), and most types of age-related macular degeneration (AMD), a good quality fundus image with an adequate field-of-view is critical. A resolution of at least 50 pixels/° along with an imager larger than 1024 × 768 pixels are required [70], and most modern-day smartphone-cameras meet this requirement. The idea of using the smartphone camera for retinal fundus imaging was first presented in Reference [69]. In this work, the image of the retinal fundus was captured with a smartphone camera through a 20 Diopter (D) lens, and a pen torch was used for illumination. Although the first of its kind, the system was not user-friendly and cannot ensure good image quality since multiple tasks such as holding the pen torchlight, the smartphone and the lens altogether while directing the light, focusing the camera, and finally pressing the touchscreen for capturing the image, must be performed. An alternative method is proposed in Reference [71], for capturing good quality fundus image. Unlike Reference [69], this system exploited the embedded flashlight of the smartphone as a light source for the camera. The user can use one hand to control the smartphone for focusing, magnifying and recording the video image while directing the light to the patient’s retina by holding a 20D or 28D ophthalmic lens with the other. The still image of the retinal fundus was then obtained from the video sequence. In Reference [72], the authors demonstrated capturing fundus images from human and rabbit eyes using a similar approach as [71]. They exploited an inexpensive application (Filmic Pro, Cinegenix LLC, Seattle, WA, USA; http://fimicpro.com/) to control the illumination, focus, and light exposure while recording the video image. Although all systems reported capturing a fundus image, no validation or comparison in reference to the standard systems was reported. Furthermore, these techniques require the use of both hands, thus leaving no room for eye indentation. In addition, the potential for peripheral retinal imaging was not explored. 

In Reference [73], a 3D printed lightweight attachment was designed for smartphones to capture high-quality fundus images. Unlike the systems reported in References [69,71,72], the attachment can hold an ophthalmic lens at a recommended but an adjustable distance from the lens of the smartphone camera and can use either the phone’s built-in flash or external light source for illumination. A similar attachment for smartphones and an external LED light source were used in Reference [68] for near visual acuity testing and fundus imaging, thus enabling remote screening of DR patients. A similar fixture was also presented in Reference [74] which has slots to accommodate a smartphone and hold a 20 D lens, thus allowing the physicians to use the system in one hand and keep the other hand free for indentation. Along with central fundus images, both systems [73,74] enables capturing high-quality images of the peripheral retina such as *ora serrata* and *pars plana*. Thus, they can be useful in screening for peripheral lesions as well, although the researchers in Reference [73] did not explore this possibility.

In Reference [75], a 3D printed plastic attachment (14 × 15.25 × 9 cm^3^) designed for the smartphones to enable high-quality wide-field imaging of retinal fundus was developed. The attachment houses three white LEDs for illumination, optical components including one 54D lens, one 20 mm focal length achromatic lens for light collection, two polarizers and a beam splitter, and a phone holder to ensure proper alignment between the camera sensor and the imaging optics. The smartphone can adjust the illumination level to the retina by regulating a battery powered on/off dimming circuit that independently controls each LED. The variability in axial length and the refractive error in the subject’s eyes were corrected by exploiting the auto-focus mechanism of the smartphone. The Ocular CellScope, as they named the complete imaging system, cost only $883 and was reportedly capable of capturing a wide field-of-view (~55°) in a single fundus image with a dilated pupil, thus making it a promising low-cost alternative to most commercial retinal fundus imaging systems. 

Similar to the Ocular CellScope, a much smaller (47 × 18 × 10 mm^3^) and less expensive ($400) magnetic attachment called D-Eye was reported in References [76] and [77]. The device houses similar optical components except for the light source, for which D-Eye exploits the smartphone’s embedded flashlight. The cross-polarization technique significantly reduced the corneal Purkinje reflections, thus making it possible to easily visualize the optic disc and screen patients, particularly for glaucoma even with undilated pupils. The authors reported achieving a field of view of ∼20° for each fundus image with a significant agreement with dilated retinal bio-microscopy which is conventionally used for grading the severity level of diabetic retinopathy. Furthermore, the retinal fundus imaging of babies was convenient due to their spontaneous attraction to the non-disturbing light emitted by the device. In addition, the device allows the examiner to work at an ergonomically convenient distance (> ~1 cm) while using the smartphone’s screen to focus the light on the patient’s retinal fundus. A stitching algorithm can be used to create a composite image, thus increasing the field-of-view of the retinal fundus. Figure 7 shows the typical arrangement of the optical components for fundus imaging with a smartphone.

A portable eye examination kit (PEEK) was presented in References [78,79] that exploited the camera of a smartphone to realize a portable ophthalmic imaging system. The system can perform an automatic cataract test by acquiring the image of an eye with a smartphone camera and comparing it to a set of preloaded images of cataract affected eyes with different intensity. However, additional external hardware is required to make PEEK suitable for fundus imaging. The additional hardware primarily comprises a red LED to illuminate the blood vessels and hemorrhages, a blue LED to observe corneal abrasions and ulcers after staining the eye with fluorescein, and a lens to magnify the image. The image of the retinal fundus is then displayed on the screen of the smartphone for diagnostic purpose. The system was successfully used to identify visual impairment in children [80,81] and to monitor the effect of prolonged exposure to extreme environments such as the long frigid darkness of Antarctica during the winter on the health of the explorers’ eyes [82]. 

### 3.4. Skin Health Monitoring

Skin cancer is one of the most common of all human cancers that is caused by the abnormal growth of skin tissue. Experts have identified three major types of skin cancer basal cell carcinoma (BCC), squamous cell carcinoma (SCC), and melanoma, the malignant form of the latter being the most dangerous one among the three. Melanoma is primarily caused by over-exposer to harmful ultra-violet rays of the sun that hinders melanin synthesis by damaging the genetic material of the melanin-producing melanocyte cells of the skin and thus putting people at high risk of skin cancer. Malignant melanoma tends to spread to other parts of the body and may turn fatal if not diagnosed and treated early. According to the America Cancer Society, 91,270 new cases of skin melanoma were estimated to be diagnosed in the US in 2018, out of which 9320 deaths were estimated [83]. Skin cancer is characterized by the development of precancerous lesions with varying shape, size, color and texture. Apart from skin cancer, there are other types of skin diseases such as psoriasis, eczema, and moles that require medical attention, thus causing loss of productivity and increase in medical expenditures. In fact, one in four Americans underwent treatment for at least one skin disease in 2013, costing $11 billion in lost productivity and $75 billion for treatment and associated costs [84]. Therefore, a low-cost solution for early detection of skin disease would be of immense use. Smartphones, being widely popular at present, can offer a viable solution for early diagnosis of skin diseases, most particularly for remote screening and long-term monitoring of skin lesions. Figure 8 shows some common medical conditions associated with the skin.

To evaluate the efficacy of smartphone-based imaging in assessing the evolution of skin lesions, a cross-sectional study of skin disease was conducted in Reference [85]. Photographs were taken and sent by the patients themselves reportedly assisted the physicians to strengthen, modify and confirm the diagnosis in 76.5% of the patients, thus significantly influencing the diagnosis and treatment of skin diseases. Furthermore, in comparison to the conventional paper-based referral system, it was observed that smartphone-based Teledermoscopy (TDS) referral schemes can significantly reduce the waiting time for skin-cancer patients [86]. In addition, the quality of most images captured by the smartphones was good enough to reliably improve the triage decisions, thus potentially making the management of patients with skin cancer faster and more efficient as compared to the traditional paper-based referrals.

Most existing portable solutions for skin disease detection rely on conventional image processing techniques along with conventional monochrome or RGB color imaging [87,88]. However, owing to its poor spatial and spectral resolution, conventional imaging approaches may not be suitable for resolving heterogeneous skin lesions [89,90] and can potentially lead to diagnostic inaccuracies. The spectral imaging techniques, on the other hand, exploits the spectral reflectance characteristics of affected sites on the skin [91,92] to resolve heterogeneous lesions and was reported to be useful in differentiating skin diseases, such as early-stage melanoma from dysplastic nevi and melanocytic nevi [93], and various acne lesion types [94]. A ring-shaped light source for the smartphone camera to study the feasibility of using smartphones for skin chromophore mapping was presented in Reference [95]. The ring comprises one white LED and three LEDs with different emission spectrum (red, green and blue) as well as two orthogonally oriented polarizers. The polarizers reduce the specular-reflected light from the skin, thus allowing the camera sensor to detect light scattered only from the skin [96]. The white light enables identifying the skin malformation, while the distribution of skin chromophores is estimated from the reflected or scattered RGB lights. Thee different smartphones were tested both in vitro and in vivo to estimate the Hemoglobin index (HI) and the Melanin index (MI). Both HI and MI reportedly increased in the in-vitro test with the concentration of absorbents in the phantoms. In vivo tests further showed the Hemangiomas and the Nevi having higher HI and MI, respectively compared to the healthy skin. Although the results achieved from all the smartphones were promising, they lacked consistency to some extent, which was attributed to the built-in automatic camera settings such as white balance, and ISO of some smartphones, and the difference in spectral sensitivities of the cameras. 

A smartphone-based miniaturized (92 × 89 × 51 mm^3^) multi-spectral imaging (MSI ) system, which is somewhat similar in principle to the approach in Reference [95] was implemented in Reference [90] to facilitate portable and low-cost means of skin disease diagnosis. Instead of using four different LEDs as in Reference [95], the system exploits the smartphone’s flashlight, nine narrow-band band-pass filters, and a motorized filter wheel to periodically obtain light of different wavelengths, which along with a plano-concave lens, a mirror, a 10× magnifying lens, and two linear polarizers are housed in a compact enclosure. An Android application (SpectroVision) was developed to control a custom interface circuit, perform multispectral imaging and analyze skin lesions. By controlling the motorized filter remotely via Bluetooth, the application captures nine images at different wavelengths ranging from 440 nm to 690 nm and one white-light image. The captured images are sent to a skin diagnosis/management platform over WiFi or LTE, where further processing of the images such as gray-scale conversion, shading correction, calibration and normalization is performed. A quantitative analysis of the nevus regions using the system was performed. In addition, quantifying acne lesions using ratio-metric spectral imaging and analysis was reported, thus demonstrating the potential of the system in diagnosing and managing skin lesions.

In order to capture the cellular details of human skin with a smartphone, a low-cost and first-of-its-kind confocal microscope was developed and used in Reference [97]. A two-dimensional confocal image of the skin was obtained by using a slit aperture and a diffraction grating, thus avoiding any additional beam scanning devices. The diffraction grating spread the focused illumination line on different locations on the tissue with different wavelengths. The authors reported observing characteristic cellular structures of human skin, including spinous and basal keratinocytes and papillary dermis with a lateral and axial resolution of 2 μm and 5 μm, respectively. However, in comparison to commercial systems, the designed system has a shallow imaging depth, which can be attributed to the use of slit aperture and a shorter wavelength (595 nm) light source of the smartphone device. In addition, the low illumination efficiency limits the imaging speed to only 4.3 fps. A light source of longer wavelength and enhanced coupling with the illumination slit [98] may improve the imaging depth and speed. Nevertheless, the small dimension (16 × 18 × 19 cm^3^) and low cost ($4200) of such a high-resolution imaging system may enable the system to be used in rural areas with limited resources for standard histopathologic analysis.

A smartphone-based system named DERMA/care was proposed in Reference [99] to assist in the screening of melanoma. An inexpensive and small dimension (6 cm × 4 cm × 2 cm) off-the-shelf microscope was mounted on the camera of the smartphone for capturing high-resolution images of the skin. In addition, a mobile application that can extract some characteristic features from the image of the affected skin was developed. The application was used to extract textural features such as entropy, contrast and variance, and geometric features such as area, perimeter and diameter. The features were then fed to a support vector machine (SVM) classifier to enable automatic classification of skin lesions. In another work, a multi-layer perceptron (MLP) was employed on a smartphone to analyze skin images captured by its camera, thus enabling skin cancer detection [100]. A similar application was proposed in Reference [101], which upon capturing the image of the skin, can alert the users about a potential sunburn and/or the severity of melanoma. There a novel method to compute the time-to-skin-burn by utilizing the information of burn frequency level and UV index level was introduced. Additionally, for dermoscopic image analysis, a system for the smartphones that incorporates algorithms for image acquisition, hair detection and exclusion, lesion segmentation, feature extraction, and classification was developed. After excluding the hair and identifying the ROI, a comprehensive set of features were extracted to feed to a two-level classifier. The authors reported achieving high accuracy (>95%) in classifying among benign, atypical, and melanoma images.

### 3.5. Mental Health Assessment

As mentioned earlier, present-day smartphones have a number of embedded sensors such as accelerometer, GPS, light sensor and microphone. The data from these embedded sensors can be collected passively with the smartphone, which coupled with the user’s phone usage information such as call history, SMS pattern and application usage may potentially be used to digitally phenotype an individual’s behavior and assess one’s mental health. For example, an individual’s stress level or emotional state can be deduced from their voice while talking over the phone and recording the conversation with the smartphone’s microphone [102,103,104]. In addition, the accelerometer can provide information about physical activity and movement during sleep. The GPS can provide information about the location and thus the context and variety of activity. Therefore, the smartphone enables a less intrusive and more precise alternative to the traditional self-reporting approach, and it may be very useful in assessing the mental wellbeing of an individual.

Many researchers used the smartphone data to assess or predict an individual’s general mental health such as social anxiety [105], mood [106,107] or daily stress level [108]. The GPS location data of 16 university students were analyzed in Reference [105], and it was reported that there was a significant negative correlation between time spent at religious locations and social anxiety. The accelerometer data along with device activity, call history and SMS patterns were analyzed in References [106] and [107] to predict mood. A prediction model based on the Markov-chain Monte Carlo method was developed in Reference [107] and it achieved an accuracy of 70% in mood prediction. In Reference [106], personalized linear regression was used to predict mood from the smartphone data. In Reference [108], by extracting device usage information and data from smartphone sensors (accelerometer, GPS, light sensor, microphone), an attempt was made to determine the factors associated with daily stress levels and mental health status [108]. There, researchers found a correlation between sleep duration and mobility with the daily stress levels. They also found speech duration, geospatial activity, sleep duration and kinesthetic activity to be associated with mental health status.

Some works in the literature also exploited the sensor data and usage information of the smartphone to assess specific mental health conditions such as depression [109,110,111,112,113,114], bipolar disorder [115,116,117,118,119], schizophrenia [119,120,121,122] and autism [123]. A significant correlation between some features of the GPS location information and depression symptom was observed in Reference [111]. For example, in Reference [114], the authors analyzed the data from the smartphone’s GPS, accelerometer, light sensor and microphone as well as call history, application usage, and SMS patterns of 48 university students. They observed that the students’ depression was significantly but negatively correlated with sleep and conversation frequency and duration. Some researchers [109,110,112,113] attempted to predict depression from the smartphone sensor data. In Reference [109], they predicted depression based on the accelerometer, GPS and light sensor data from the smartphone. Both the support vector machine (SVM) and random forest classifier were used in Reference [113] to predict depression from the GPS and accelerometer data, as well as from the calendar, call history, device activity and SMS patterns. However, the prediction accuracy in References [109] and [113] was only slightly better than the chance. In Reference [110], only the GPS data was used to predict depression and with the SVM classifier, moderate sensitivity and specificity of prediction were achieved. In Reference [112], both the GPS data and the device activity information were exploited to predict depressive symptoms. Using a logistic regression classifier for prediction, the authors achieved a prediction accuracy of 86%.

Some significant correlations between the activity levels and bipolar states were observed in some individual patients, where the physical activity level was measured with the smartphone’s accelerometer [115]. In Reference [119], the authors used data from the Bluetooth, GPS sensor and battery consumption information of the smartphone to track an individual’s social interactions and activities. There, they found that the data from these sensors are significantly correlated with their depressive and manic symptoms. In References [116,117], the accelerometer and GPS data of the smartphone were used to detect the mental state and state change of persons with bipolar disorder. They reported detecting state change with 96% precision and 94% recall and achieved an accuracy of 80% in state recognition [116]. Using additional information from the microphone and call logs, the precision and recall increased to 97%; thus improving the reliability of the system. However, the state recognition accuracy was somewhat reduced, which was attributed to the noisy ground-truth and inconsistencies in the daily behavioral patterns of the participants [117]. In Reference [118], the accelerometer and light sensor data of the smartphone, the call history and SMS patterns were exploited in a generalized and a personalized model to predict the state among persons with bipolar disorder. A precision and recall of 85% and 86%, respectively was achieved in state prediction.

In Reference [120], a study on the feasibility and acceptance of passive sensing by smartphone sensors among the people with schizophrenia was conducted. Persons with schizophrenia were mostly found open to sensing with smartphones and two-thirds expressed interest in receiving feedback, but a third expressed concern about privacy. The GPS location information was exploited to recognize outdoor activities among people with schizophrenia and thereby to infer social functioning [121]. In Reference [122], the authors proposed a system called CrossCheck that used data from the GPS, accelerometer, light sensor and microphone as well as call history, application usage, and SMS patterns to predict the change in mental health among patients with schizophrenia. There, they collected data from 21 patients and observed statistically significant associations between the patients’ mental health status and features corresponding to sleep, mobility, conversations and smartphone usage. The authors used random forest regression to predict the mental health indicators in the patients with schizophrenia and reported achieving a mean error of 7.6% with respect to the scores derived from the participants’ responses to a questionnaire.

Recently, Apple Inc.’s ResearchKit initiative launched a mobile application called “Autism and Beyond” [123]. This application captures images of the users’ facial expressions in response to standardized stimuli by the iPhone’s front-facing camera and analyzes these images using algorithms designed for emotion recognition. This application can potentially identify individuals who are at risk of autism and other developmental disorders. A large-scale trial is currently underway to assess the validity and utility of this approach. 

### 3.6. Activity and Sleep Monitoring Systems

Daily physical activities such as walking, running and climbing stairs involve several joints and muscles of the body and require proper coordination between the nervous system and the musculoskeletal system. Therefore, any abnormalities in the functioning of these biological systems may potentially affect the natural patterns of these activities. For example, persons at the early onset of Parkinson’s disease tend to exhibit small and shuffled steps, and occasionally experience difficulties to start, stop and take turns while walking [22,124]. Additionally, due to gradual deterioration of motor control with age, older adults are at high risk of fall and mobility disability. In fact, an estimated 10% (2.7 million) of Canadians, aged 15 years and over, suffered from mobility-related disabilities in 2017 [125]. Furthermore, falls in the older adults may cause hip and bone fractures, joint injuries, and traumatic brain injury, which not only require longer recovery time but also restrict physical movement thereby affecting an individual’s daily activities. In addition, fall-related fractures reportedly have a strong correlation with mortality [124]. Moreover, nearly one-third (30%) of Canadian adults between 18 and 79 years of age were estimated to be at intermediate or high risk for sleep apnea [126], which is often associated with high blood pressure, heart failure, diabetes, stroke, attention deficit/hyperactivity disorder, and increased automobile accidents [127,128]. Therefore, quantitative assessment of gait, knee joints and daily activities including sleep are critical in early diagnosing musculoskeletal or cognitive diseases, sleep disorders, fall and balance assessment, as well as in the post-injury rehabilitation period.

Most existing activity monitoring systems rely on a network of cameras fixed at key locations in a home [22,129]. Although such systems can provide comprehensive information about complex gait activities, they are expensive and generally have a limited field-of-view. In recent years, there has been a growing interest in using smartphone embedded motion sensors such as accelerometers, gyroscopes, and magnetometers as well as location sensors such as the GPS sensor for real-time monitoring of human gait and activities of daily living (ADL) [130,131,132,133,134,135,136,137,138,139,140,141,142,143,144,145,146,147,148,149,150,151,152,153,154,155,156,157]. These sensors measure the linear and angular movement of the body, and the location of the user, which can be used to quantify and classify human gait events and activities in real time. A general architecture of smartphone-based activity monitoring is presented in Figure 9.

At the heart of an activity monitoring and recognition system is the classification or recognition algorithm. However, signal processing techniques and extraction of appropriate features also play critical roles in realizing a computationally efficient and reliable system. Signal processing techniques may include filtering, data normalization and/or data windowing or segmentation. Subsequently, a good number of key features from the statistical, temporal, spatial and frequency domains are extracted to feed into the classification model. Table 3 presents a list of typical features that are extracted from the motion signals. Finally, an appropriate classification model such as support vector machine (SVM) [137,138,155,158,159], naive Bayes (NB) [136,149,155,156], k-means clustering [138,149], logistic regression [134,155,156], k-nearest neighbor (KNN) [133,136,155,156,158], neural network (NN) [140,141,143,151,152] or a combination of models [134,140,141,148,150] are employed for activity recognition.

The Manhattan distance metric was used in Reference [130] to compare the accelerometer data of an average gait cycle from a test sample to three template cycles corresponding to three different walking speeds. The authors attempted both statistical and machine learning approaches and the highest accuracy (~99%) in classifying three different walking speeds was achieved with the support vector machine (SVM). An important limitation of this approach is that it relies on the local peak and valley detection to identify the gait cycles, but their consistency varies with walking speed and/or style. A two-stage continuous hidden Markov model (CHMM) was proposed in Reference [131] for recognition of human activities. Some subsets of optimal features were first produced by employing the random forest importance measures. The static and dynamic activities were then distinguished by applying the first-level CHMM, which was followed by a second-level CHMM for achieving a finer classification of the activities with an accuracy of ~92%. In Reference [132], a fuzzy min-max (FMM) neural network based incremental classification approach was used to learn activities, which includes walking, ascending and descending stairs, sitting, standing, and laying. The authors then applied a classification and regression tree (CART) algorithm to predict these activities and reported achieving a recognition accuracy of ~96.5%. A voting scheme was adopted in Reference [134] to combine the classification results from an ensemble of classifiers such as a J48 decision tree, logistic regression (LR) and multilayer perceptron (MLP). These authors reported identifying four activities such as walking, jogging, sitting and standing with an accuracy of more than 97%. However, the proposed approach performed poorly in distinguishing between activities like ascending and descending stairs, where the recognition accuracy reduced to ~85% and ~73%, respectively. In Reference [135], four smartphones were attached to the waist, back, leg, and wrist and captured motion data from the accelerometer and gyroscope for activity measurement; humidity, temperature, and barometric pressure sensors for sensing environmental parameters; and Bluetooth beacons for location estimation. A modified conditional random field (CRF) algorithm was implemented on each unit to classify the activities individually using a set of suitable features extracted from the preprocessed sensor data. The decisions from each unit were then assessed based on their relevance to the body positions to finally determine the activities. The authors reported identifying 19 daily activities including cooking, cleaning utensils and using bathroom sink and refrigerator with more than 80% accuracy.

An orientation independent activity recognition system based on smartphone embedded inertial sensors was reported in Reference [137]. The raw sensor data were processed by signal processing techniques including coordinate transformation and principal component analysis. A set of statistical features were extracted from the processed sensor signals, which is then fed to several classification algorithms such as ANN, KNN and SVM to identify the same six activities (walking, ascending and descending stairs, sitting, standing, and laying) investigated in References [131,132,133]. The authors also presented an online-independent SVM (OISVM) for incremental learning that can deal with the inherent differences among the measured signals resulted from the variability associated with the device placement and the participants. There they reported identifying the activities with an accuracy of ~89% using OISVM. In order to deal with the high computational costs associated with the machine learning techniques for activity recognition, a hardware-friendly support vector machine (HF-SVM) based on fixed-point arithmetic was proposed in Reference [138]. There the authors reported achieving a recognition accuracy of 89%, which is comparable to the performance of the conventional SVM.

Some researchers [139,140,141,142,143,150,151,152] exploited more advanced techniques such as deep neural networks for activity recognition. Unlike conventional machine learning approaches, these techniques do not employ separate feature extraction and feature selection schemes. Rather, they automatically learn the features and perform activity recognition simultaneously. In References [140,141], a deep convolutional neural network (Convnet) was formed by stacking several convolutional and pooling layers to extract key features from the raw sensor data. The Convnet, being coupled with multilayer perceptron (MLP) can classify six activities as [131,132,133] with an accuracy of ~95%. By incorporating additional features extracted from the temporal fast Fourier transform (tFFT) of the raw data, the performance of the Convnet was found to improve by ~1%. The authors in Reference [142] implemented an activity recognition system using a bidirectional long short-term memory (BLSTM)-based incremental learning approach that exploits both the vertical and horizontal components of the preprocessed sensor data to obtain a two-dimensional feature. Several such BLSTM classifiers were then combined to form a multicolumn BLSTM (MBLSTM). The authors compared the performance of MBLSTM with that of several classifiers such as SVM, kNN and BLSTM, and achieved the lowest error rate (~15%) in recognizing seven activities that include jumping, running, normal and quick walking, step walking, ascending and descending stairs. In Reference [150], a smartphone-based activity recognition system was developed using a deep belief network (DBN). There the authors first extracted a set of five hundred and sixty-one features from the motion signals following a signal processing step that includes signal filtering and data windowing. A kernel principal component analysis (KPCA) was then employed on these features and only the first one hundred principal components were fed into the DBN while training the model for activity recognition. The authors reported achieving an accuracy of ~96% in recognizing 12 activities that include standing, sitting, lying down, walking, ascending and descending stairs, stand-to-sit, sit-to-stand, sit-to-lie, lie-to-sit, stand-to-lie, and lie-to-stand. However, unlike most deep learning-based system, the approach in Reference [150] requires a stand-alone feature extraction step, thus increasing the computational load. In general, the key issues associated with the deep learning-based approaches are their high computational cost, making them unsuitable for real-time applications, especially for devices with limited high-end processing capabilities. 

Some researchers exploited smartphones for fall detection [144,145,157,158,159,160,161,162] and posture monitoring [146,147]. In Reference [144], the angle between the longitudinal axis of the device and the gravitational vector was continuously monitored using the smartphone embedded motion sensor. When this angle drops below a pre-determined threshold of 40°, the system recognizes the event as a fall. The proposed system, when worn on the waist, was able to distinguish a fall from a normal body motion, i.e., lying, sitting, static standing and horizontal/vertical activities with high accuracy. However, the system requires the phone to remain attached to the waist for fall detection, which may not be comfortable or always feasible for users. A similar fall detection system was implemented in Reference [145] where the smartphone was kept in the shirt pocket with its front side facing the body in order to maintain consistency in the orientation of the sensors. The system monitors the changes in the acceleration along the three directions and detects a fall if the change happens faster than an experimentally established minimum time spent on performing normal activities of daily living. However, no quantitative information regarding the accuracy of the system was provided in both References [144,145]. In Reference [159], the authors proposed an SVM-based fall detection algorithm. They used smartphone’s built-in accelerometer to record user’s motion data by placing the phone in the front pocket of the shirt. The participants performed some activities of daily living (ADL), and also some simulated fall events. The authors reported distinguishing fall events from non-fall activities with high sensitivity (~97%) and specificity (95%). A detailed review of smartphone-based fall detection systems was presented in References [161,162].

In Reference [146], the authors reported a smartphone-based posture monitoring application named Smart Pose. The accelerometer data and facial images were captured simultaneously while holding the smartphone and facing towards it. The pitch and roll orientation of the device, and thereby the cumulative average of tilt angle (CATA) of the user’s neck is estimated, assuming the device’s orientation correlates to and represents the position of the user’s neck. A bad posture is determined when CATA exceeds an acceptable range of 80° to 100°. The authors compared the performance of their proposed system with a commercial three-dimensional posture analysis system and reported achieving similar performance with the Smart Pose. An application named iBalance-ABF was proposed in Reference [147] to assess the balance of the body and to provide audio feedback to the user accordingly. The smartphone embedded accelerometer, gyroscope and magnetometer were used to determine the tilt of the mediolateral trunk. When the tilt angle of the trunk exceeds an adjustable but predetermined threshold, the system sends audio feedback to the user over the earphone. The smartphone, mounted on a belt, however, needs to be attached to the back at the level of the L5 vertebra, which requires assistance. Nevertheless, iBalance-ABF can be useful, particularly for older adults to improve their posture and balance.

Applications based-on smartphone-sensors that facilitate monitoring of knee joints [162,163], sleeping patterns [164,165,166] and sleep disorders [167] were reported. Table 4 summarizes several recent smartphone-sensor based activity and sleep monitoring systems. In addition, it should be noted that there are published research results that used external inertial measurement units (IMUs) for activity recognition [7,22,142]. For example, the authors in Reference [143] combined the convolutional neural network (CNN) and LSTM to form a deep convolutional LSTM (DeepConvLSTM) and used this approach for activity recognition.The CNN can determine the key features from the signal automatically, while the LSTM translates the temporal patterns of the signals into features. The authors reported recognizing a complex set of daily activities with high precision (F1 score > 0.93). Smartphone embedded motion sensors such as accelerometers, gyroscopes and magnetometers can replace these external IMUs and achieve similar performance. However, in order to recognize a complex set of activities, as it was reported in Reference [143], it requires several sets of IMUs/smartphones to be attached at different parts of the body, which is not cost-effective and may not be practical. 

### 3.7. Hearing Impairment Monitoring Systems

According to the World Health Organization (WHO), 6.1% of the world’s population including one-third of the adults aged 65 or above suffer from different levels of hearing loss [168]. Moreover, the number of people with disabling hearing impairment is estimated to grow rapidly over the coming years, reaching 630 million by 2030 and over 900 million in 2050 [168]. Hearing impairment at mild to moderate level may cause people to lose 50%–70% of speech in the noisy environment [169,170,171] and can degrade a person’s quality of life, if not corrected [171,172]. Generally, a hearing test is performed by the pure-tone audiometry (PTA) to determine the hearing threshold levels of an individual based on the patients’ feedbacks to pure tones stimuli. Therefore, PTA is usually recommended for patients over five years of age i.e., old enough to follow the test procedures [173]. In addition, PTA requires trained personnel, special infrastructure and arrangements to keep the ambient noise levels low during the test and regular maintenance of the systems to ensure high precision and accuracy in the test results [174,175]. Consequently, hearing care services are expensive and so are scarce, particularly in developing and lesser-developed countries. Fortunately, smartphone-based hearing assessments applications can provide a low-cost and faster alternative to conventional hearing screening procedures.

In Reference [176], a smartphone application for hearing loss screening called uHear was evaluated in twenty-six subjects aged 84.4 ± 6.7 years. The result was compared with respect to that obtained from a standard portable audiometer as well as from the participants’ feedback obtained through a questionnaire. For most frequencies, the pure tone thresholds for the uHear application were found to be higher than the audiometric thresholds (40 dB), which was attributed by the authors to the poor quality of the earbuds, causing leakage of audio signals and penetration of ambient noise to the ear canal. However, 92% of the test results from the application were found to be in agreement with that obtained from the audiometer, resulting in a screening sensitivity and specificity of 100% and 60%, respectively. Another such application, HearScreen™ was investigated in References [177,178] for assessing hearing loss. In Reference [177], the HearScreen™ application was operated by community health workers (CHW) to perform community-based screening of hearing loss through home visits over a period of 12 weeks. The referral rates by HearScreen™ were reported to be 12% and 6.5% for children (2–15 years) and adults (16–85 years), respectively. Although the authors in Reference [177] reported receiving positive feedbacks from the CHWs about the application in terms of its usability, screening time, and community need, the sensitivity and specificity of the application were not verified. However, the performance of HearScreen™ was investigated in Reference [178] with respect to a clinical audiometer. There, they investigated HearScreen™ application to screen for hearing loss among 1,236 participants and validated the test results with clinical pure-tone audiometry. With HearScreen™, the authors reported achieving high sensitivity (81.7%) and specificity (83.1%) in screening for hearing loss at an average screening time of around one minute.

A smartphone-based audiometer was presented in Reference [179] that used custom designed hearing aids as the audio source. There, they implemented a program on the smartphone that controlled the hearing aids over a low-power wireless communication medium to generate audio stimuli. The generated audio signals showed a little variation of less than 0.5 dB hearing level (HL) at all six frequencies (1 kHz, 2 kHz, 4 kHz, 8 kHz, 500 Hz and then 250 Hz) compared to the expected values. The system was further tested on twenty subjects with different degrees of hearing loss in a sound-proof environment and verified with respect to a conventional audiometer. While assessing hearing-loss severity at different levels, the proposed system demonstrated little difference (<6 dB HL on average) in the hearing thresholds with respect to the conventional audiometer. Owing to the systems significantly reduced (50%) screening time, high portability and low cost in comparison to the conventional audiometers, the system can, therefore, be very useful and effective for hearing loss screening at point-of-care diagnostics in rural and urban areas.

A detailed review of some ear and hearing assessment applications was presented in Reference [180]. Only a small subset of all these applications was reported to have been investigated by peer-reviewed studies and the reported performance of these applications in terms of screening accuracy, referral rates, sensitivity and specificity varied across the studies. Nevertheless, even though a smartphone based system may not determine the degree of hearing loss as accurately as a conventional audiometer, it may be useful and effective for faster initial screening for hearing loss at home or at primary healthcare centers in the rural and urban areas of developing and least-developed countries. 

To assist hearing-impaired people, hearing aids are generally prescribed by the physicians. Hearing aids amplify the audio signals entering the ears, thus improving the audibility of the sound. Smartphone-based hearing aids can allow the users to control the volume and frequency-gain response as per their comfort level, thereby making them a viable alternative to conventional hearing aids. In Reference [181], the feasibility of smartphone applications based hearing aids was studied on a group of people aged between 50 and 90 years with mild-to-moderately severe hearing loss, who had been using hearing aids for less than 3 months. The participants used a conventional hearing aid and two smartphone apps, EARs and microphone, each for 2–3 weeks. While using the smartphone apps they attached an inline microphone to the shirt and in-ear headphones to the phone. From the electroacoustic measurements and speech-in-noise test, the authors observed similar performance among all three devices. However, the authors attributed the differences in the placement of microphones in the hearing aid compared to the smartphone apps for similar speech-in-noise performance. In addition, the participants overall showed greater satisfaction with smartphone applications compared to the hearing aids.

A smartphone-based hearing assistive system was presented in Reference [182] to assist people with mild-to-moderate hearing loss. The application (SmartHear), picks up the speaker’s voice by the smartphone’s microphone, converts the analog signal to pulse-code-modulated (PCM) digital signals. The PCM signals are then transmitted over the Bluetooth medium to a receiver, which converts the digital signal back to the analog domain and sends the analog signal to the ear through headphones. The authors reported achieving an average improvement in speech intelligibility by 0.2 on a scale of 0–1 at four different noise levels and across four different audiograms for mild-to-moderate hearing loss. However, unlike the traditional hearing aids, the smartphone’s microphone in the proposed architecture resides near the speaker’s vicinity, which may not be always feasible for practical use.

## 4. Regulatory Policies

As smartphone-based health monitoring systems and applications are increasing rapidly and becoming more pervasive in society, there is a growing concern about the safety issues and associated potential dangers [183,184]. Concerns also remain among many researchers about whether and/or how a regulatory policy would be adopted and enforced by the government bodies such as the US Food and Drug Administration (FDA), and Medicines and Healthcare Products Regulatory Agency (MHRA) in the United Kingdom (UK) [185,186].

Many experts [187,188,189,190,191,192,193,194,195] have raised questions about the accuracy and reliability of smartphone-based health monitoring applications/systems, the vast majority of which reportedly lacked enough involvement of medical professionals during the design and evaluation phases. For example, in Reference [189], the authors studied and tested a dermatology app called ‘Skin Scan’, which was found to recognize only 10.8% images correctly as high-risk melanomas against 93 clinical images from the National Cancer Institute and Fitzpatrick’s Dermatology in General Medicine. Furthermore, in January 2017, a team of researchers [196] conducted a search for suitable apps in the iTunes App Store and Google Play that can assist people to deal with anxiety disorders and selected 52 apps for study. They found that 63.5% (33 out of 52) of the apps were reported as having no information about the intervention approach. In addition, no information related to the manufacturers’ professional credentials were available for more than two-thirds (35 out of 52) of the applications. Only two out of the 52 anxiety apps were found to be thoroughly tested by the psychiatrists [196]. Therefore, cautious use of many of these applications was advised in References [190,192] due to their diagnostic inaccuracies and unreliability. Some of them are reported to be unsafe to use [188,193,194] and even may cause life-threating consequences [195]. Hence, adoption and enforcement of some regulatory policies were recommended by the experts [187,191] to ensure accountability, data-privacy, information security and patient welfare in terms of safety and diagnostic effectiveness.

Following the FDA’s release of a draft guideline for regulating mobile medical apps in 2011, key experts in this industry expressed their expectations regarding the policies for medical apps [197]. These experts urged the FDA to draw a clear demarcation line between the medical apps and the fitness or wellness app, as well as between diagnosing apps and monitoring apps. They recommended for defining the risk-level threshold of regulatory significance for medical apps. They also suggested defining the boundaries of FDA regulations for apps serving as device accessories and making a guideline to deal with the modular applications. In February 2015, the FDA released the latest version of the guidelines defining the categories of smartphone-based healthcare applications that must require regulatory oversight [198]. According to the guideline, FDA will regulate only those medical applications that can turn a smartphone into a medical device such as ophthalmoscopes and dermatoscopes using external and/or internal sensors and devices. Regulatory oversight will also be applied to those applications that can be used as an accessory to the FDA-approved medical devices such as a smartphone-based ECG monitor. In short, regulatory oversight from the FDA is required if any application that can possibly affect the ‘performance or functionality’ of the FDA-regulated medical devices, and thus may pose a risk to patient safety. However, some mobile applications, although being a medical device by the definition, enjoy “enforcement discretion” as they pose a low risk to patients [198]. Regardless, all high-risk class III devices and about 75% of medium-risk Class II devices require clinical trials and/or other evidence to demonstrate their safety and compliance with the intended operation [184]. However, if a device demonstrates substantially similar performance to an already approved and legally marketed device (predicate device), it may enjoy an exemption from new clinical trials upon proper evidence that shows the device has same intended use and technological characteristics as the predicate device [199]. In the case when the new device has different technological characteristics in terms of device safety and effectiveness—first, it must not raise any new concerns, and second, it must meet the minimum standards of the predicate device [199]. Nevertheless, there remain serious concerns about safety assurances in the process of device approval based on predicate devices. Therefore, both the Institute of Medicine [200] and the U.S. Congress [201] understandably urged to curtail this approach of device approval. So far, the FDA has approved several healthcare applications developed for the mobile platform [202]. The diagnostic radiology app ‘Mobile MIM’ is the first such application ever available in iTunes stores [203]. This app allows a healthcare professional to view, assess and securely share images with patients, peers or partner institutions, thus reducing diagnosis and treatment delay. KardiaMobile (AliveCor, Inc.) is another FDA approved device that comes with an application, which can turn a smartphone into a portable single-lead electrocardiogram (ECG) machine [204]. Other FDA approved healthcare apps for smartphones include the iExaminer™ (Welch Allyn, Inc.) adapter for PanOptic™ Ophthalmoscope [205], BlueStar^®^ (WellDoc, Inc.) for type 2 diabetes management [206], and ResolutionMD^®^ (PureWeb Inc., Calgary, Canada) for viewing and assessing diagnostic images [207].

In Europe, according to the EU Medical Device Directive MDD 93/42/EEC [208], published on 14 June 1993 and amended in the Directive 2007/47/EC [209], any stand-alone or combination of ‘instrument, apparatus, appliance, software, material or other article’ intended for healthcare purposes including diagnosis, monitoring, prevention and treatment will be considered be as ‘medical device’. Therefore, most smartphone-based healthcare applications including those that monitor and assess cardiovascular health, eye and skin health through imaging, and lung health, will fall under the umbrella of ‘medical device’ and thus require Conformité Européenne (CE) certification for marketing the product within the European Economic Area (EEA). The CE certification or ‘CE marking’ ensures the product’s conformity with the health, safety, and environmental protection standards set by the EU’s harmonization legislation [210]. For example, an Irish app ONCOassist™ [211] was designed as a decision support tool for professional oncologists at the point-of-care and incorporated prognostic tools, drug interaction checker, survival rate predictors for diseases such as breast cancer, colon cancer, and lung cancer. In addition, this app also incorporated some algorithms to determine, for example, liver cirrhosis severity, level of consciousness, the prognostic score for patients with advanced Hodgkin lymphoma and appropriate dosage of chemotherapy agents based on patient’s body surface area (BSA) and thereby, was considered as a medical device. ONCOassist™ received the CE certification in 2013 and displays the CE mark on its welcome screen. 

A new medical device regulation (MDR) [212] (EU) 2017/745 was published in the Official Journal of the European Union repealing the MDD 93/42/EEC on 5 May 2017. The new MDR brings previously unregulated non-medical and cosmetic devices under the umbrella of ‘medical device’, with many of them being reclassified as medium to high risk (such as class IIa, IIb and III) devices. The other key changes in the MDR over the MDD includes inclusion of medical purpose devices and active implantable medical devices (AIMD), requirements for the manufacturers to update clinical data, technical documentation, and labeling; and generate and provide detailed clinical data to validate safety and performance claims and enforce unique device identification (UDI) for tracking. A wider range of smartphone-based commercial healthcare apps will now be defined as the ‘medical devices’ according to the MDR that require the manufacturers, and app development companies to revisit their safety and quality control processes to ensure compliance with the new MDR, that is scheduled to be enforced on May 26, 2020. Although these changes are meant to ensure a much safer, transparent and sustainable regulatory framework for the consumer, changeover on such a scale in a limited timeframe is a mammoth task for manufacturers and regulatory bodies to achieve. Furthermore, the scheduled parting of the United Kingdom (UK) from the EU—popularly termed as ‘Brexit’—is causing more confusion for the manufactures to this already highly challenging task. It was unclear whether the UK would comply with EU regulations [213]. However, the UK Government, on 4 July 2017, vowed to work closely with the EU in terms of medicines regulation to ensure public health and safety even after leaving the EU [214]. In a recent statement, the UK’s Department of Health and Social Care declared that it will comply with the key elements of the MDR and recognize all medical devices approved for the EU market and CE-marked after leaving the EU, in the case a no Brexit deal is reached [215].

Currently, the Medicines and Healthcare Products Regulatory Agency (MHRA) of the UK complies with the existing Medical Device Directive (MDD) and defines any healthcare app or software as a ‘medical device’ based on the functionality or service it provides to the users and the associated risks in terms of patient’s safety [216]. According to the MHRA, an app/software is most likely to be considered as a ‘medical device’ if it is designed to perform some calculations or run some algorithms on the raw data to detect, diagnose and prevent disease, or to monitor the course of a disease or injury. Apps that are intended for archiving records without modification, providing existing information, and making general recommendations for an expert’s advice, can safely be excluded from ‘medical devices’ category. However, if the decision-support apps perform some calculations or interpret or interpolate the data and do not allow the clinicians to review the raw data, then such apps/software are highly likely to fall into the ‘medical device’ category. Apps that perform simple and straightforward calculations to track physical fitness such as heart rate, step-count or BMI (body mass index) are not considered as ‘medical devices’. However, apps/software that perform complex calculations, for example, to determine medicine doses can potentially fall into the high-risk class III ‘medical device’ category [217,218,219]. The MHRA recommends the users to use a CE marked medical purpose app to ensure user safety. 

In order to receive a ‘CE mark’ for the medical purpose apps—a ‘medical device’ by the MDD 93/42/EEC—the manufacturers need to identify the class of the device based on the perceived risk associated with it and select the corresponding conformity assessment procedure. The conformity assessment procedure ensures tighter control to be applied to the device in case the perceived risks associated with it is higher. Next, the manufacturer prepares a document that generally includes the technical details about the design and manufacturing process of the device as well as the intended operation of the product to demonstrate the product’s compliance with the MDD 93/42/EEC. For a low-risk i.e., class I device, the manufacturer can self-declare the device’s compliance with the Directive. For class IIa devices, manufacturers must also declare the device’s compliance with the corresponding regulatory requirements of the Directive. Additionally, class IIa devices as well as class IIb and class III devices must require a notified body (NB) to carry out a detailed conformity assessment and receive a ‘Declaration of Conformity’ certificate from the NB to submit as an evidence of the app/software’s being compliant with the MDD 93/42/EEC [208].

However, it was argued in a report to the U.S. Congress of the Global Legal Research Center that the ‘CE mark’ on a medical device does not necessarily ensure the quality of the device in terms of its performance and clinical effectiveness, rather it merely shows its compliance with the EU legislation [220]. Medical devices in the EU are approved based on the safety and performance standards, and demonstration of the devices’ clinical efficacy is not required by the MDD [184]. However, the new MDR, which is scheduled to be in force in 2020 has put more emphasis on clinical trials and evidence [212]. On the other hand, the US Food and Drug Administration (FDA) requires the devices to ensure not only the safety and performance but also their clinical efficacy [221]. Furthermore, only one organization, the FDA, governs the entire process of device approval in the US. While this ensures better surveillance on the regulatory processes, however, often it turns out to be an expensive, rigid and lengthy process for manufacturers [222]. In contrast, in the EU, the manufacturers can flexibly appoint one of the many EU approved private, for-profit ‘notified bodies’ to assess and approve the devices in terms of regulatory standards, thus expediting the process for obtaining a ‘CE’ mark, but this may be, at a potential risk of compromised safety [223,224]. In addition, some ’high-risk medical devices developed, for instance, by an academic institution, can likely be distributed through/among the associated entities for non-commercial use without a CE mark, whereas in the US, prior FDA-approval is an absolute necessity before distribution [225]. However, the approval process of medical devices based on the predicate device and without rigorous new clinical evidence can deter the manufactures to carry out expensive and time-consuming clinical trials, which not only raise concerns in terms of device safety and efficacy but also may lessen the scope of device improvement and innovation [225].

While the US and the EU represent 40% of the global markets for medical devices [226], other markets such as Canada, Australia, and Japan have their own regulatory bodies to enforce regulatory policies for medical devices. Health Canada, for example, categorizes the medical devices into four classes from Class I to Class IV based on the risks associated with the devices [227]. Prior to marketing a medical device in Canada, the manufactures or the distributors must apply for and receive the Canadian Medical Device License (MDL) for class II, III, and IV devices and the Medical Device Establishment License (MDEL) for class I devices [227]. However, the information required to file an application for Health Canada approval is approximately the same as that required in the US and EU [228]. On the other hand, Australia’s Therapeutic Goods Administration (TGA) relies mostly on the EU regulations and CE mark certification from the European NBs before granting approval to market medical devices there [229]. Recently, Australia’s TGA decided to begin recognizing registrations and certifications from additional foreign medical device regulators including US FDA, Health Canada, the Japanese Pharmaceutical and Medical Devices Agency (PMDA) [230].

In 2014, the International Medical Device Regulators Forum (IMDRF) launched the Medical Device Single Audit Program (MDSAP) pilot to develop an efficient and standardized global directive to auditing and monitoring medical devices [231]. The regulatory bodies participating in this program include TGA of Australia, Agência Nacional de Vigilância Sanitária (ANVISA) of Brazil, Health Canada, the U.S. FDA, and the Ministry of Health and Labor and Welfare (MHLW) of Japan, while the EU participated as an observer [231]. TGA has recently decided to recognize registrations and certifications from MDSAP auditing organizations [230]. Starting in January 2019, Health Canada also planned to discard the Canadian Medical Device Conformity Assessment System (CMDCAS) and replace it with the MDSAP certification. In fact, Health Canada urged the MDL holders to submit evidence for MDSAP transition from CMDCAS and/or MDSAP certificates by the 31st December 2018 [232].

## 5. Conclusions and Research Challenges

In this paper, we have presented a state-of-the-art survey on health and activity monitoring systems that exploit the embedded sensors in smartphones for measuring physiological parameters and tracking health conditions. The ubiquity of smartphones has grown enormously in the past decade. In addition, the significant advances in sensor technologies in terms of size, cost, energy requirements and sensitivity has enabled the integration of a number of sensors into present-day smartphones. The embedded sensors in smartphones such as the image sensor, microphone, ambient light sensor and motion sensors coupled with modern high-speed data transfer technologies may assist people to lead an independent and active life while ensuring non-invasive monitoring of their health and physical well-being in a regular fashion without adding much to their personal expenses.

Monitoring the health of the heart, eye, respiratory systems and skin, as well as the activities of daily living (ADL) and mental conditions in a continual fashion, can provide detailed information about an individual’s overall health and wellbeing over a prolonged period of time. The smartphone and its embedded sensors coupled with present-day information and communications technologies have opened a new window of opportunity for cost-effective remote healthcare services. The raw medical data thus obtained by the smartphone sensors can be sent over the internet to a remote healthcare facility for detailed investigation. Furthermore, the incredible improvements in the processing and data storage capabilities in the modern-day smartphones may allow for faster, real-time and onboard execution of complex predictive algorithms and/or artificial intelligence (AI) technologies using the high-volume of raw data measured by the smartphone sensors. Thus, smartphones may play an incredible role in enabling a low-cost solution for early diagnosis through continuous monitoring, initial screening of diseases such as melanoma, and diabetic retinopathy and remote monitoring of the progression of some diseases.

Owing to the high market penetration and ever-increasing computational capabilities of smartphones, there have been growing interests among the researchers and the manufacturers in building smartphone-based devices for healthcare and wellness purposes. However, there remain some key challenges that need to be addressed prior to achieving a global acceptance of smartphones as medical devices.

First, most of the works reported in the literature are based-on nonrandomized, non-blinded studies on a limited number of subjects using a proof-of-concept device. Further, the limited size and possible bias in the samples implies that the universal efficacy of the devices is still a critical concern. Therefore, rigorous clinical trials are required to evaluate the safety and efficacy of the proposed smartphone-based ‘medical’ devices.

Second, although major regulatory bodies have their own guidelines for a medical app to be considered as a ‘medical device’, the boundaries between the fitness and wellness apps and the medical apps remain ambiguous, particularly in a situation when the self-monitoring thorough a fitness app is integrated within the patient care and treatment scheme. These blurred boundaries need to be resolved to safeguard the users from possible harmful consequences.

Third, unlike the US Food and Drug Administration (FDA), the intermediate- and high-risk devices in the EU require an authorized private and for-profit third-party organization called the ‘Notified Body’ (NB) to assess and certify the device’s compliance with the corresponding directive. Although this process offers more flexibility to the manufacturers and reduces unnecessary delay in the approval process, it is, however, subject to the risk of compromised safety owing to varying standards, pricing and work ethics of different NBs. 

Fourth, approving a medical device based on a predicate may cause safety concerns and was therefore criticized by some experts. It was argued that some predicates were in the market even before any regulatory policies were implemented. Some predicate devices were never tested on humans and some were even recalled voluntarily from the market due to their poor performance, thus questioning the credibility of the predicate itself. In addition, this process of device approval encourages the manufacturers to evade the expensive and time-consuming but critical clinical trials before bringing the product in the market. 

Fifth, among the abundant number of healthcare and wellness apps published to-date, the user is often left perplexed in finding and using the most appropriate and safe one, specific to his/her needs. A centralized database or a dedicated app store of approved medical apps can be of immense benefit for both product developers and consumers. The centralized system, similar to other app stores, can serve as a common platform for both the users and the developers. It can also review and recommend the apps based on the quality, reliability, medical effectiveness, safety, privacy and value-for-money.

Sixth, in order to ensure widespread acceptance of the smartphone-based health monitoring devices/apps among the users, these devices/apps need to be affordable, easy-to-use, and compatible with the most mobile operating systems as well as smartphones from different manufacturers. Therefore, more research and development efforts are needed to improve the systems’ ease-of-use and pervasiveness. 

Seventh, one of the major concerns for smartphone-based healthcare systems is associated with the privacy and security of any sensitive medical information. However, to date, most publications either did not address these critical issues or did so in a cursory manner. Therefore, more efforts are needed to develop and implement robust algorithms to ensure data privacy and information security.

## Figures and Tables

**Figure 1 sensors-19-02164-f001:**
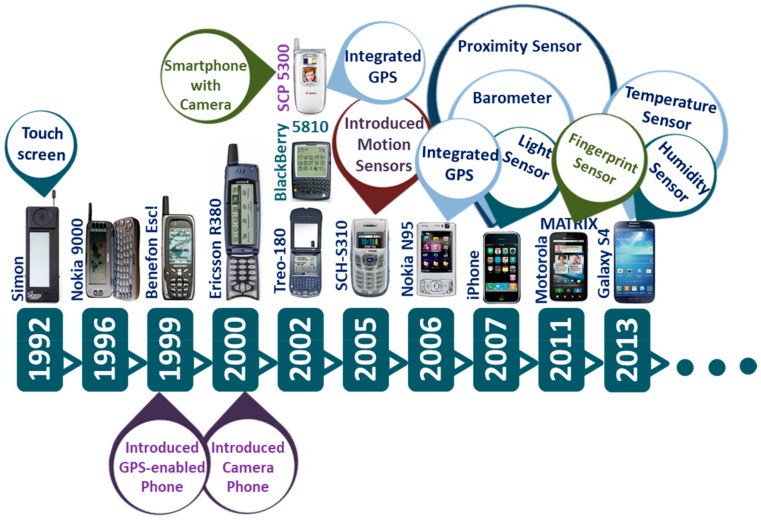
Evolution of smartphones and smartphone-embedded sensors over time.

**Figure 2 sensors-19-02164-f002:**
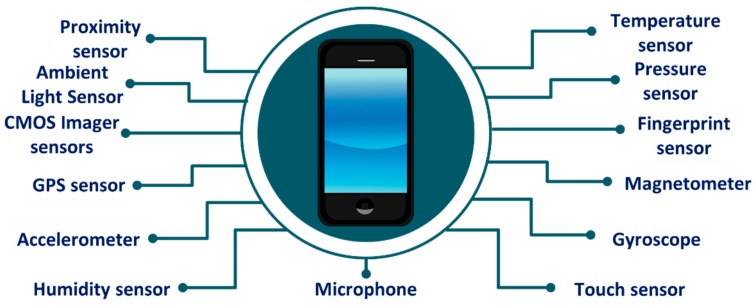
Built-in sensors in a typical present-day smartphone.

**Figure 3 sensors-19-02164-f003:**
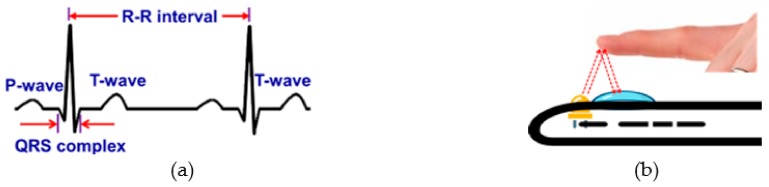
Measuring heart rate (**a**) from a typical trace of a single lead Electrocardiogram (ECG) signal, and (**b**) using a smartphone camera.

**Figure 4 sensors-19-02164-f004:**
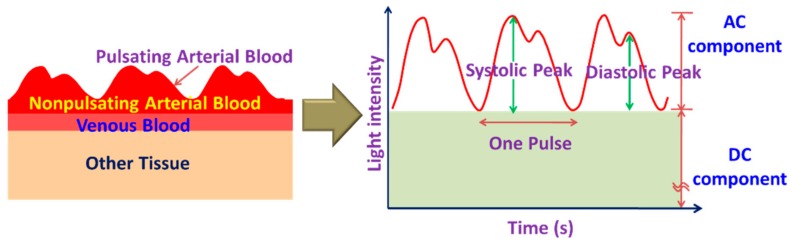
Photoplethysmograph (PPG) signal obtained from the pulsatile flow of blood volume.

**Figure 5 sensors-19-02164-f005:**
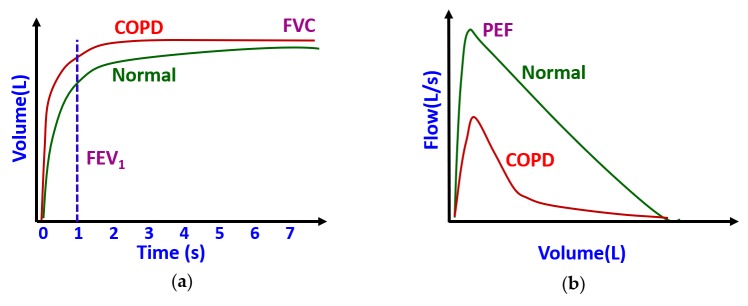
Typical spirometric flow curves (**a**) volume-time curve, and (**b**) flow-volume curve.

**Figure 6 sensors-19-02164-f006:**
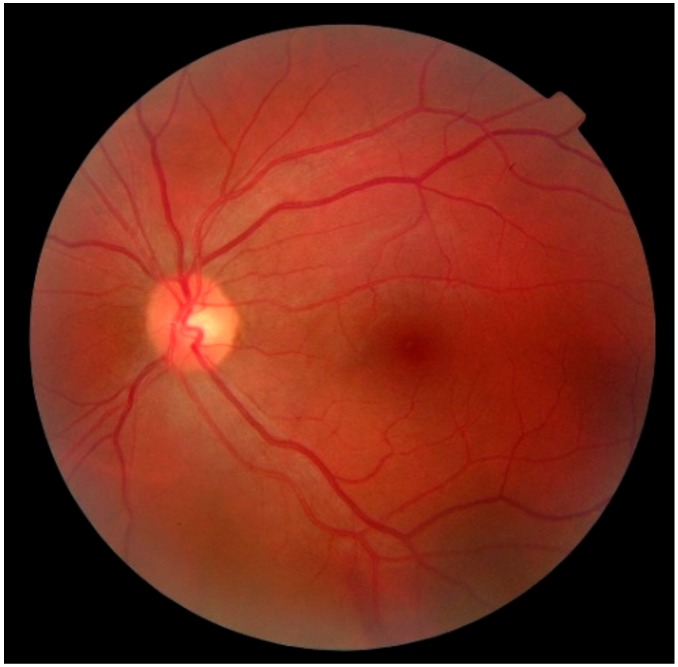
Image of the retinal fundus of a healthy eye; Source: https://pixabay.com/en/eye-fundus-close-1636542/.

**Figure 7 sensors-19-02164-f007:**
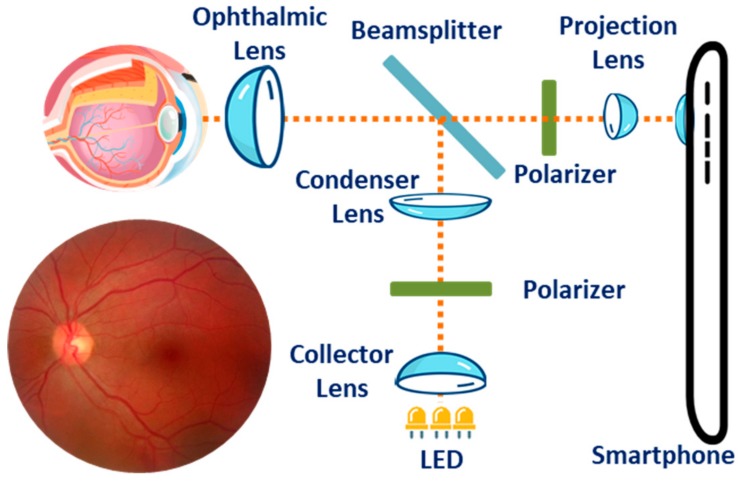
Typical arrangement of the optical components for fundus imaging with a smartphone.

**Figure 8 sensors-19-02164-f008:**
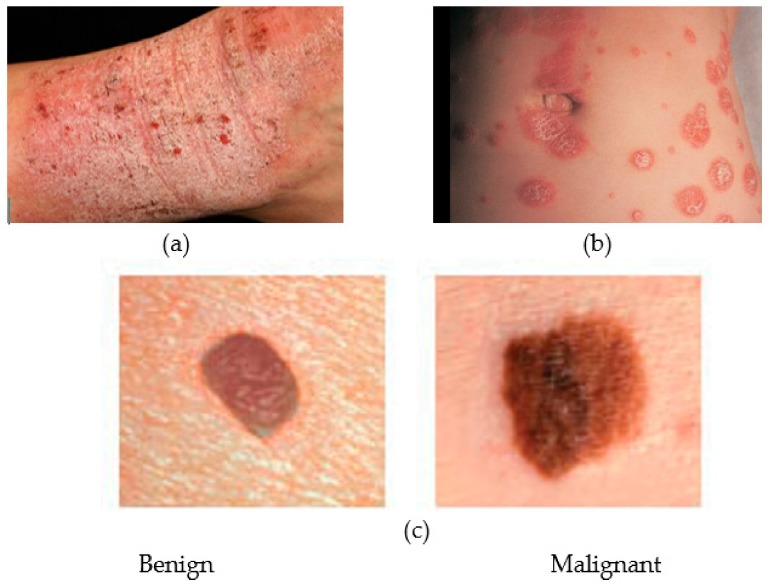
Several types of skin diseases (**a**) Eczema, (**b**) Psoriasis and (**c**) two forms of Melanoma.

**Figure 9 sensors-19-02164-f009:**
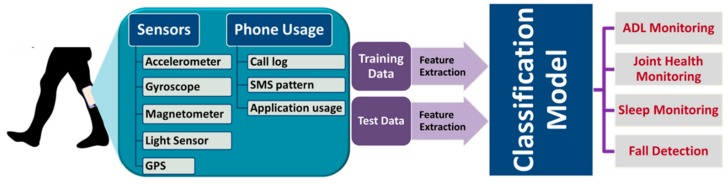
General architecture of a smartphone-based activity monitoring system.

**Table 1 sensors-19-02164-t001:** Smartphone sensors used for health monitoring.

Monitored Health Issues	Typically Used Smartphone Sensors
Cardiovascular activity e.g., heart rate (HR) and HR variability (HRV)	Image sensor (camera), microphone
Eye health	Image sensor (camera)
Respiratory and lung health	Image sensor (camera), microphone
Skin health	Image sensor camera)
Daily activity and fall	Motion sensors (accelerometer, gyroscope, proximity sensor), Global positioning system (GPS)
Sleep	Motion sensors (accelerometer, gyroscope)
Ear health	Microphone
Cognitive function and mental health	Motion sensors (accelerometer, gyroscope), camera, light sensor, GPS

**Table 2 sensors-19-02164-t002:** Smartphone-sensors for cardiovascular health monitoring.

Ref.	Year	Measured Signs	Type	Smartphone Model	Sensor Used	Video Resolution	Frame Rate (fps)	Video Length	Method	Performance *wrt* Standard Monitors	# of Subjects
[37]	2018	HR, HRV	Contact-based (index finger)	iPhone 6, Apple Inc., Cupertino CA	Front camera	1280 × 720	240	5 min	• Reflection of light from the finger is measured.	Pearson Correlation coefficient (PC) for most parameters between PPG and ECG: >0.99	50 (11 F, 39 M)
[39]	2016	HR, HRV	Contact-based (index finger)	iPhone 4S, Apple Inc., Cupertino CA	Rear camera		30	5 min	• Combination of the steepest slope detection of pulse wave derived from the green channel of the reflected light and its correlation to an optimized pulse wave pattern.	PC: >0.99 (HR), ≥0.90 (HRV)	68 (28 F, 40 M)
[38]	2016	HR, RR	Contact-based (HR) and contactless (RR)	HTC One M8, HTC Corporation, New Taipei City, Taiwan	Front (for RR) and rear (for HR) camera	RR: 320 × 240 (ROI: 49 × 90 abdomen) HR: 176 × 144 (ROI:176 × 72)	30 (down-sampled to 20 (RR), 25 (HR))	--	• Frequency domain analysis of the noncontact video recordings of chest and abdominal motion.	Average of median errors for RR: 1.43%–1.62% between 6 and 60 breaths per minute	11 (2 F, 9 M)
[45]	2012	HR	Contactless (face)	iPhone 4, Apple Inc., Cupertino CA	Front camera	640 × 480	30	20 s	• Analysis of the raw video signal (green channel) and ICA-decomposed signals of the face in the frequency domain.	Error rate: 1.1% (raw signal), 1.5% (ICA-decomposed signals)	10 (2 F, 8 M)
[49]	2018	HR, RR	Contactless (face)	LG G2, LG Electronics Inc., Korea	Rear camera	--	30 (down-sampled to 10)	20 s	• Frequency domain analysis of the color variations in the reflected light (hue) from the face.	PC: 0.9201 (HR) and 0.6575 (RR)	25 (10 F, 15 M)
[36]	2016	HR	Contact-based (index finger)	--	Rear camera	1920 × 1080	--	--	• Frame-difference based motion detection for improving data quality. • Uses all 3 channels (R, G, B) for PPG extraction.		20
									• Blood volume flow was observed clearly in the Red channel.	Average accuracy: 98%	
[50]	2015	Pulse, HR, HRV	Contact-based (index finger)	Motorola Moto X, Motorola, Libertyville, IL and Samsung S 5	Rear camera	640 × 480	30	100 s	• Extracts PPG by averaging the Green channel data of the video. • HR is calculated by detecting the consecutive PPG peaks.	PC of pulse and R-R interval from two phone models > 0.95	11
[40]	2014	HR, NPV	Contact-based (index finger)	iPhone 4S, Apple Inc., Cupertino CA	Rear camera	ROI: 192 × 144	30	20 s	• HR and NPV were measured in the presence of a controlled motion (6 Hz) of the left hand. • Evaluated the effect of motion artifact (MA) on the PPG in all three color (R, G, B) channels.	Higher SNR for B and G channel PPG in presence of 6Hz MA. PC: HR>0.996 (R, B, G), NPV = 0.79 (G)	12 (M)
[51]	2014	HR, HRV	Contact-based (index finger)	Sony Xperia S, Sony Corporation, Tokyo, Japan.	Rear camera	--	--	60 s	• HR was estimated by detecting the consecutive PPG peaks and also the dominant frequency. • Combines several parameters (HR, HRV, Shannon entropy) to detect Atrial fibrillation (AF).	HR error rate: 4.8% AF detection: 97% specificity, 75% sensitivity	
[52]	2012	HR, HRV	Contact-based (index finger)	iPhone 4s and Motorola Droid, Motorola, Libertyville, IL	Rear camera	ROI: 50 × 50	30 (iPhone), 20 (Droid)	2, 5 min (iPhone, Droid)	• Several ECG parameters were extracted with two different models of smartphone both in supine and tilt position and performed comparative analysis with the data obtained from a standard five lead ECG.	PC: ~ 1.0 (HR), PC for Other ECG parameters: 0.72-1 (Droid), 0.8-1 (iPhone)	9 (iPhone) 13 (Droid)
[44]	2012	HR	Contact-based (index finger)	HTC HD2 and Samsung Galaxy S	Rear camera	ROI: 288 × 352 (HTC) 480 × 720 (Samsung)	25 (HTC) 30 (Samsung)	6 s	• HR is calculated by detecting the consecutive PPG peaks.	Error: ± 2 bpm	10
[53]	2012	HR	Contact-based (index finger)	Motorola Droid, Motorola, Libertyville, IL	Rear camera	ROI: 176 × 144	20	5 min	• HR from the PPG signals was obtained at sitting, reading and video gaming by using an Android-based software.	PC: ≥ 0.99 Error: ± 2.1 bpm	14 (11 F, 3 M)

**Table 3 sensors-19-02164-t003:** Typical features extracted from motion signals [22].

Spatial Domain	Temporal Domain	Frequency Domain	Statistical Domain
Step length	Double support time	Spectral power	Correlation
Stride length	Stance time	Peak frequency	Mean
Step width	Swing time	Maximum spectral amplitude	standard deviation
RMS acceleration	Step time		Covariance
Walking speed	Stride time		energy
Signal vector magnitude (SMV)	Cadence (steps/min)		Kurtosis

**Table 4 sensors-19-02164-t004:** Smartphone-sensor based activity monitoring systems.

Ref.	Proposition	Phone	Sensors	Experiment Protocol	n	Method	Performance/Comment
[130]	Human activity and gait recognition	Samsung Nexus S	a, ω	• Subjects walked ~30 m for each of three different walking speeds • Smartphone in the trouser pocket • Sampling rate: 150 sample/s	25	• Each gait cycle was detected and normalized in length. • Several distance metrics between the test and template cycle were calculated as features. • Statistical analysis and machine learning used for recognition.	• Gait recognition accuracy 89.3% with dynamic time warping (DTW) distance metric. • Activity recognition accuracy >99%.
[131]	Human activity recognition	Samsung Galaxy S II	a, ω	• University of California Irvine (UCI) Human activity recognition (HAR) dataset • Subjects performed an activity twice, with the phone (1) mounted on the belt at the left side (2) placed according to the user’s preference.	30	• Feature selection using random forests variable importance measures. • Two-stage continuous HMM for activity recognition. • First and 2nd level for coarse classification and fine classification, respectively.	• Activity (walking, ascending and descending stairs, sitting, standing, and laying) recognition accuracy 91.76%.
[132]	Human activity recognition	Samsung Galaxy S II	a, ω	• UCI HAR dataset • Activities are: walking, ascending and descending stairs, sitting, standing, and laying	30	• A hybrid model based on the fuzzy min-max (FMM) neural network and the classification and regression tree (CART).	• Activity (walking, ascending and descending stairs, sitting, standing, and laying) recognition accuracy 96.52%.
[133]	Evaluation of hyperbox (HB) NN for classifying activities	Samsung Galaxy S II	a, ω	• UCI HAR dataset • Five subsets of varying sizes (5%, 10%, 20%, 50% and 100% of the dataset) were created for training purpose	30	• One HB is assigned for all attributes of a class and has one or more associated neurons for class distribution. • Points falling into (1) only one HB are immediately classified (2) overlapping regions of HBs use the neural outputs for prediction.	• Performance was comparable to SVM, decision tree, KNN and MLP classifier. • Activity (walking, ascending and descending stairs, sitting, standing, and laying) recognition accuracy 75%–87.4%
[134]	Human activity recognition	Nexus One, HTC Hero, Motorola Backflip	a	• Wireless sensor data mining (WISDM) dataset from http://www.cis.fordham.edu/wisdm/dataset.php • Sampling rate: 20 samples/s	36	• Extracted 43 features from the mean and standard deviation of acceleration, mean absolute difference, mean resultant acceleration, time between peaks and binned distribution. • A Voting scheme to combine the results from the J48 decision tree, logistic regression and MLP.	• Accuracy > ~97% (walking, jogging, sitting and standing), ~86% (ascending stairs), and ~73% (descending stairs)
[156]	Human activity recognition	iPod Touch	a, ω	• Measured activities: sitting, walking, jogging, and ascending and descending stairs at different paces	16	• Evaluated different classification models (decision tree, multilayer perception, Naive Bayes, logistic regression, KNN and meta-algorithms such as boosting and bagging) in terms of recognition accuracy.	• Accuracy for sitting, walking, and jogging at different paces: 90.1%–94.1% • Accuracy for ascending and descending stairs: 52.3%–79.4%
[135]	Complex activity recognition system	Samsung Galaxy S IV	a, ω, P, T, H ( and Gimbal beacons)	• Four smartphones worn on the waist lower back, thigh, and wrist. • Participants performed 19 activities in 45 minutes according to their own order of choice and repetition.		• Conditional random field (CRF) based classification was performed on each device separately.	• Activity recognition accuracy > 80%
						• Final recognition was based on the result from the most relevant device to that particular activity. • 19 activities are: walk and run indoors, clean utensil, cook, sit and eat, use - bathroom sink and refrigerator, move from/to indoor to/from outdoor, ascending and descending stairs, stand, lie on the - bed, floor, and, sofa, sit on the bed, floor, sofa, and, toilet.	
[136]	A feature selection approach for faster recognition	Samsung Galaxy S II	a, ω	• UCI HAR dataset • Activities are: walking, ascending and descending stairs, sitting, standing, and laying.	30	• Data segmentation by sliding window and extraction of time and frequency domain features • A hybrid of the filter and the wrapper (FW) methods for feature selection • Performance verified by naïve Bayes and KNN.	• Activity recognition Accuracy, precision and F1-score to 87.8%, 88.0% and 87.7% (with a, ω data) • Significant reduction in recognition time.
[137]	Algorithm for Human activity recognition	Google NEXUS 4	a, ω	• Subjects performed each activity twice for 30 s each, keeping the device at five different orientations.	5	• Employed coordinate transformation and principal component analysis (CT-PCA) on the data to eliminate the effect of orientation variation. • Used several classification models for evaluation.	• Activity (static, walking, running, going upstairs, and going downstairs) recognition accuracy 88.74% with online-independent SVM (OISVM)
[138]	A hardware friendly SVM for HAR	Samsung Galaxy S II	a, ω	• UCI HAR dataset • Activities are: walking, ascending and descending stairs, sitting, standing, and laying.	30	• Standard support vector machine (SVM) with fixed-point arithmetic for computational cost reduction.	Activity recognition accuracy ~89% (similar to standard SVM)
[139]	Unsupervised learning for activity recognition	Samsung Galaxy Nexus	a, ω	• Smartphone was kept in a pants pocket for measurements • *n* = 5 activities: walking, running, sitting, standing, and lying down • Each activity was performed for 10 min.	--	• Experiment 1: known *n*. k-means, Gaussian mixer model, and average-linkage hierarchical agglomerative clustering (HIER) were used for recognition. • Experiment 2: unknown *n*. Density-based spatial clustering of applications with noise (DBSCAN) along with three other models used for classification.	• GMM achieved 100% recognition accuracy when *n* is known • HIER and DBSCAN achieved over 90% recognition accuracy when *n* is unknown.
						• DBSCAN requires setting two parameters (*eps* and *minPts*) and for other models, *n* was chosen based on local maxima of the Calin´ ski–Harabasz index (CH).	
[140,141]	DNN for Human activity recognition	Samsung Galaxy S II	a, ω	• UCI HAR dataset • Activities are: walking, ascending and descending stairs, sitting, standing, and laying.	30	• DNN was formed by stacking several convolutional and pooling layers to extract discriminative features.	
						• Number of layers, number of feature maps, pooling and convolutional filter size were adjusted to maximize test-accuracy by ‘softmax’ classifier.	
						• Multilayer perceptron for final recognition.	• Accuracy: 94.79%–95.75%
[148]	Human activity recognition	Samsung Galaxy S II and Huawei P20 Pro	a, ω	• Smartphone was attached to the waist. • Sampling frequency = 50 Hz • 10,299 samples with Samsung Galaxy SII and 4752 samples with Huawei P20	30	• An Ensemble Extreme learning machine with Gaussian random projection (GRP). • GRP was used for the initialization of input weights of base ELMs.	Activity (sitting, standing, laying, walking, walking upstairs and downstairs) recognition accuracies: 97.35% (Samsung), 98.88% (Huawei)
[154]	Human activity recognition	Samsung Galaxy Note I, Motorola Droid,		• Collected 2 weeks of GPS data continuously • Subjects prepared a journal of real-time information about their everyday activities.	3	• A fuzzy logic -based approach for classification.	Classification accuracy: ~96%
						• Location uncertainty improved by calculating the probabilities of different activities at a single location.	
		Nokia N900	GPS			• Recognized activities by a segment aggregation method while adjusting for location uncertainties.	
[149]	Human activity recognition	Samsung Galaxy S 4	a, ω	• Free walk at a natural pace and run in a straight path, maintain a standing position and minimize additional bodily movement (25 s each).	1	• Feature set consisted of linear acceleration, normal acceleration and angular velocity. • Naive Bayes and k-means clustering for classification	Classification accuracy: 85%
[150]	Human activity recognition		a, ω	• A database of 12 activities (standing, sitting, lying down, walking, ascending and descending stairs, stand-to/from-sit, sit-to/from-lie, stand-to/from-lie, and lie-to/from-stand).	--	• Extracted features were processed by a kernel principal component analysis (KPCA) and linear discriminant analysis (LDA).	
						• Deep belief network (DBN) for classification.	Mean recognition rate: 89.61% and overall accuracy: 95.85%
[151]	Human activity recognition	Huawei Mate 9	a, ω	• Activities were logged approximately 5–8 hours a day for 4 months	1	• A six-layer independently recurrent neural network (IndRNN) processed data of different lengths and captured the temporal patterns at different time intervals.	Classification accuracy: ~96%
[152]	Human activity recognition	Samsung Galaxy S II	a, ω, ф, and P	• UCI HAR dataset • Activities are: walking, ascending and descending stairs, sitting, standing, and laying	30	• DNN-based subassembly divides sensor data into various motion states. The transformation subassembly derives the intrinsic correlation between the sensor data and personal health.	• Accuracy: 95.9% with unsupervised feature extraction • 96.5% with manual feature extraction
[153]	Walk@Work(W@W)-App for HAR	--	a, ω	• 1 h laboratory protocol and two continuous hours of occupational free-living activities	17 (10F 7 M)	• Calculated agreement, intra-class correlation coefficients (ICC) and mean differences of sitting time against the inclinometer ActivPAL3TM, and step counts against the SW200 Yamax Digi-Walker pedometer for performance comparison.	• ICC: 0.85 for self-paced walking, 0.80 for active working tasks. • ICC (free-living): 0.99, 0.92 with a difference of 0.5 min and 18 steps for sitting time and stepping, respectively.
[155]	Human activity recognition	Samsung Galaxy S II	a, ω, ф,	• Four smartphones attached to four body position: right pocket, belt, right arm, and right wrist • Measured activities: walking, running, sitting, standing, walking upstairs and downstairs	4	• Data from three types of sensors were evaluated in terms of recognition accuracy using seven classifiers (naïve Bayes, SVM, neural networks, logistic regression, KNN, rule-based classifiers and decision trees).	• Best performance was achieved using both gyroscope and accelerometer data together. • Magnetometer data played little role.
[147]	Balance analysis and Audio Bio-Feedback (ABF) system	iPhone 4	a, ω, ф, mic	• Smartphone was mounted on a belt. • Subjects wore the belt on the posterior low back at the level of the L5 vertebra and a pair of earphones, placed arms close to the trunk, stood barefoot, with their eyes closed.	20 (11F and 9 M)	• Tilt angles and heading were calculated from accelerometer and gyroscope, respectively as well as from the magnetometer.	--
						• Kalman filter was used to correctly estimate the rotation angles from the difference between the two previous estimates. • Audio feedback sent through the mic when trunk orientation is above a threshold.	
				• Subjects kept sway minimum in parallel feet (10 cm apart), tandem stance-positions, and 2 experimental conditions with and without ABF. • Each experimental condition was performed in random order six times, each for 30 s.			
[144]	Fall detection and notification system	Lenovo Le-phone	a	• Smartphone mounted on the waist	--	• Extracted signal magnitude area (SMA), signal magnitude vector (SMV) and tilt angle from the median filtered accelerometer data.	
						• Fall detection with a decision tree-based algorithm. • In case of a fall, a multimedia messaging service (MMS) was sent with time and location info.	• Performance comparison not reported.
[145]	Fall detection	Samsung Galaxy S III	a	• Collected acceleration data		• Detected a fall if the acceleration along a direction changed at a faster rate than that in normal daily activities.	• Performance comparison not reported.
[157]	Fall detection, tracking and notification system	--	a	• Evaluated the tracking error range at two outdoors and one indoor fall location. • Tests conducted near a school and a subway station at three periods of the day: 7:00–12:00, 12:00–18:00, and 18:00–24:00 to evaluate the accuracy of tracking with mobile obstacles.	10	• Calculated accelerometer SMV. • Rapid change in the SMV to a large value indicated a fall. • In case of a fall detected, the GPS location of the smartphone is communicated. • The real-time location tracking system used Google’s 3D mapping services.	• Overall accuracy of the location tracking system: < 9 m. • Larger error range observed between 12:00 and 18:00. • High density of Wi-Fi installations improves location accuracy.
[158]	Fall detection and daily activity recognition	Sony C6002 Xperia Z, Apple iPhone 4s	a, ω, ф	• Subjects kept phones in the right, left and front-pockets and fall onto a 15 cm thick cushion. • Activities: four types of fall (forward, backward, toward the left and right) and ADL.	8	• Activities were classified using supervised machine learning (SVM, Decision tree, KNN and discriminant analysis) algorithms. • A fall is detected when SMV goes above a threshold value (24.2 ms^-2^).	• ADL (sitting, standing, walking, laying, walking upstairs and walking downstairs) recognition accuracy 99% with the SVM.
[159]	Fall detection algorithm	Sony Z3	a	• Smartphone was placed in the front pocket • Subjects performed six activities of daily living and six fall activities	10 (7 M and 3 F	• Six features (SMV, sum vector excluding gravity magnitude, max and min value of acceleration in gravity vector direction, mean of the absolute derivation of acceleration in gravity vector direction, and gravity vector changing angle) were derived from the accelerometer data.	
						• SVM was used to classify fall and non-fall events.	• 96.67% sensitivity, 95% specificity
[160]	Fall detection based on high-level fuzzy petri net (HLFPN)	HTC Desire S	a	• Smartphone was placed in the thigh pocket • Activities: Falls (forward, backward, vertical, and sideways) and ADLs (walking, jogging, jumping, sitting, and squatting).	12 (7 F and 5 M)	• Calculated accelerometer SMV and frequency of occurrences from the accelerometer data.	• Fall detection accuracy 90% with HLFPN
						• Fuzzy degree was generated by substituting the calculated values into the membership function formulated by the experiment. • Final classification with HLFPN.	
[163]	Knee Joint ROM	iPhone 6	a	• Dynamic knee extension ROM was measured three times with an interval of 5 min. • Phone was attached to the tibia • An isokinetic dynamometer used to generate and measure the knee motion for validation.	21 (M)	• A MATLAB program automatically detected the min/max values of knee extension angles from the accelerometer data. • The difference between the min and max values was calculated as the dynamic knee extension ROM.	• Highly correlated (*r_s_* = 0.899) and low error (~0.62°) *wrt* the commercial system (Biodex System 4 Pro) • Limits of agreement: −9.1 to 8.8 deg. • ICCE between two methods >0.862
[164]	Assessment of smartphone apps for measuring knee range of motion	--	Camera, inclinometer (a, ω, ф)	• Five measurements of knee range of motion from each subject by a commercial system, two apps - Goniometer Pro and Dr. Goniometer • Goniometer Pro (by 5fuf5) and Dr. Goniometer (by CDM S.R.L.) were based on smartphone inclinometer and camera, respectively.	10 (5 F and 5 M)	• Goniometer Pro: attached to the anterior of the thigh proximal to the skin incision, and on the anterior of the distal tibia distal to the skin incision and knee flexion angle ($\theta_{Fx}$) was derived by adding the two measured angles.	• $\theta_{Fx}$ by Dr. Goniometer was clinically identical to $\theta_{Fx}$ from the commercial system, with a mean difference of <1° and 1/50 difference >3°
						• Dr. Goniometer: calculated $\theta_{Fx}$ by taking pictures from the lateral side of the operated knee with markers virtually placed at the level of the greater trochanter, the knee joint and the ankle joint.	
[165]	An app Toss ‘N’ Turn (TNT) for sleep quality monitoring	Any Android phone (version 4.0 +)	a, Mic, light sensor, screen proximity sensor	• Subjects installed TNT in the phones and kept it in the bedroom while sleeping and entered a daily sleep diary every morning. • TNT stores sensor data, data about running processes, battery and display screen state in a protected database on the phone.	27 (19 F and 8 M)	• The time-series sensor data were divided into a series of non-overlapped 10 min windows for data analysis and feature extraction. • Extracted 32, 122 and 198 features associated with sleep detection, daily sleep quality inference, and global sleep quality inference, respectively.	• Classification accuracy: 93.06% (Sleep state), 83.97% (daily sleep quality), 81.48% (overall sleep quality)
[166]	Best effort sleep (BES) model for sleep duration monitoring		Light sensor and Mic (+ Phone usage)	• BES tracked six phone usage features (total duration of phone-lock, phone-off, phone charging, phone in darkness, phone in a stationary state and phone in a silent environment) on a daily basis for one week.	8	• Model assumption: Sleep duration is a weighted linear combination of six features. • Weights are estimated using a non-negative least-squares regression.	• Sleep duration estimation error range: ± 42 min • Estimated duration is close to commercial wearable systems.
[167]	Sleep monitoring system	iPhone	a	• Subjects recorded data for at least four consecutive nights using both the ActiGraph, attached to the non-dominant wrist and the smartphone, placed close to the pillow.	13 (4 F and 9 M)	• Four sleep measures (sleep onset latency (SOL), total sleep time (TST), wake after sleep onset (WASO), and sleep efficiency (SE%)) are extracted from both systems.	• Satisfactory agreement with the ActiGraph for all sleep parameters except for the SOL.
[127]	Contactless Sleep Apnea Detection	Samsung Galaxy S4	Phone speaker and micro-phone	• The speaker transmits 18–20 kHz sound waves and the microphone senses the reflections. • Total of 296 hours of measurements • Polysomnography (simultaneously done EEG, EMG, airflow, SpO2, electrooculogram) was conducted for performance comparison.	37 (17 F and 20 M)	• Employed FMCW (frequency modulated continuous wave) transmissions to isolate reflections arriving at different times. • Mapped the human body-specific arrival times of the reflected signals to carrier frequency shift, allowing for extracting the amplitude changes due to breathing. • Reflection patterns of non-breathing movements are different than breathing movements.	• Highly correlated (correlation coefficient of 0.9957, 0.9860, and 0.9533 for central apnea, obstructive apnea and hypopnea, respectively) with the ground truth • Average error (rate of apnea and hypopnea events) < 1.9 events/h.

*a*: accelerometer, ω: gyroscope, P: pressure sensor, T: temperature sensor, H: humidity sensor, ф: magnetometer, n: number of subjects.

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
