# Peer review of "Smartphone Sensors for Health Monitoring and Diagnosis"

_sensors, 2019, doi:10.3390/s19092164_

Round 1

Reviewer 1 Report

The work presented in your paper presents a comprehensive and exhaustive review concerning the application of smartphones for health monitoring, which is a timely research subject, plenty of interest.

With the aim of helping you to enhance your paper, I propose to you some suggestions:

The title of your article is “Embedded Sensors in Smartphones for Remote Health Monitoring”. However, in my opinion, you do not provide enough technical details (technology, power consumption, performances) concerning the sensors that are embedded in the smartphones. I agree that manufacturers do not easily offer detailed information about their products, and it is not always easy to find this information elsewhere. Perhaps, in this case, the title of the paper should be simply “Smartphones for Remote Health Monitoring”.

I think that the section Regulatory Policies is very pertinent, mainly in the context of sensible topics such as the employ of general public technologies in the health domain. However, the presented information can be improved by including not only the European and American policies, but also other policies all around the world (in example, Japan, China, Canada…). In its current state, it seems that this section only offers a comparison to stablish what policy, European or American, is the best.

Finally, in Table 4, it is not cleat what does “Sub” mean. This is the same for the α, ω, and φ angles: are they the same for all presented sensors and for all smartphone models?

Author Response

Please see word file 
2019Apr27 Smartphone-sensors for Health Monitoring & Diagnosis.docx

Reviewer 2 Report

The paper is an extensive survey on the use of the (many) embedded sensors of smartphones for remote health monitoring.

It is an important and relevant topic, as properly explained by the authors, because of the evolution of demography, with a world population living longer. Indeed, in many countries, increase of life expectancy is demanding extra resources to healthcare services and alike. The topic of the paper is also clearly within the scope of Sensors.

The paper is properly organized, with a proper introduction (section 1), a thorough timeline of the smartphone evolution (section 2), description of smartphone sensors for health monitoring (section 3), regulatory policies (section 4) and conclusions (section 5). I find Section 4 - "Regulatory policies" particularly important and interesting, as this topic is often neglected in scientific publications, although it is of the utmost importance when dealing with health. 

Topics are thoroughly covered, with relevant papers and projects properly cited and described. The paper is also well written, without issues regarding the text.

Regarding the health areas reviewed by the paper and that can benefit from the embedded sensors of smartphones, an important was left out: hearing. Considering that i) hearing impairment or, at least, reduction of the hearing function is a frequent health problem of senior citizens (and of other citizens), and ii) smartphones are particularly well equipped to deal with sound, it is a mandatory area for a survey paper. Note that some commercial solutions based on smartphones/tablets already exist for assessing hearing loss. For example, companies dealing with hearing loss/impairment uses APP for assessing hearing and promote their hearing-aid devices. I've seen special (simple) kiosks that rely on tablets providing hearing assessment tests. The tests are performed through an APP and some special (of good quality) in-ear headphones, without requiring the presence of a specialist (some are done in drugstore). The hearing assessment is performed by either by the APP which can or not be connected to a remote server, with the APP used to trigger the hearing tests and collect results. 

In my opinion, a survey paper focusing on the use of embedded smartphone sensors for remote health monitoring needs to include the assessment of hearing.  So the question is: Is there any sound reason for not including hearing in the paper? 

This is the only issue why I am requesting a major review. 

Minor observation: "Sub" should be identified as "Number of subjects" in the footnote that exists in page 22.

Author Response

(The authors gave the same response as above.)

Round 2

Reviewer 2 Report

The authors have addressed all the issues pointed out in the 1st review. Congratulations to the authors for i) writing such a comprehensive paper and ii) address the issues pointed out in review 1 in a very timely manner.